EMBO
Molecular Medicine

# JAK1/2 inhibitor ruxolitinib reduces aggregates in cardiac proteinopathy

Erda Alizoti [1,2], Leonie Ewald[1,2], Simona Parretta [1,2], Jacob J K March [1], Moritz Meyer-Jens [1,2], Ellen Orthey[1], Christian Conze[3], Lucie Carrier [1,2], Jeffrey Robbins[4] & Sonia R Singh [1,2] ✉

## Abstract

**Misfolded proteins cause or contribute to a wide range of progressive diseases that are difficult to treat. Desmin-related (cardio-)myopathy (DRM), a well-studied proteinopathy, presents with progressive muscle weakness and shortened life span. Most DRM patients display intracellular accumulation of desmin and its chaperone αB-crystallin (CRYAB). Using an unbiased high-throughput screen, we found that Janus kinase 1 (JAK1) knockdown resulted in lower CRYAB[R120G] aggregates in cardiomyocytes. Here, we tested whether the JAK1/2 inhibitor ruxolitinib ameliorates the disease phenotype in CRYAB[R120G] DRM models. Ruxolitinib cleared pre-existing CRYAB[R120G] aggregates in neonatal rat cardiomyocytes and human induced-pluripotent stem cell-derived cardiomyocytes, and enhanced ubiquitin-proteasome system (UPS)-mediated degradation. Blocking UPS function and specifically knockdown of the E3 ligase ASB2 blunted the effect of ruxolitinib on CRYAB[R120G] accumulation. In DRM mice, phospho-STAT3 and JAK1 levels were higher than in non-transgenic mice, indicating pathologically active JAK-STAT signaling. Ruxolitinib treatment resulted in lower CRYAB[R120G] aggregate load and prevented cardiac dysfunction in DRM mice. Similar findings were obtained by crossing DRM mice with the cardiomyocyte-specific *Jak1* knockout, suggesting JAK1 as a therapeutic target in proteinopathy.**

**Keywords** Ruxolitinib; JAK1; Desmin; CRYAB; ASB2
**Subject Category** Cardiovascular System

## Introduction

Intracellular protein aggregates are particularly toxic to non-dividing cells with high energy demand, such as cardiomyocytes or neurons, and result in severe myopathies or neurodegenerative diseases (Chiti and Dobson, 2017). Genetic variants in desmin (*DES* (Goldfarb et al, 2008)), αB-crystallin (*CRYAB* (Vicart et al, 1998)), myotilin (*MYOT* (Selcen and Engel, 2004)), Z-band alternatively spliced PDZ-motif (*ZASP* (Griggs et al, 2007)), filamin C (*FLNC* (Brodehl et al, 2016)), BCL2-associated athanogene 3 (*BAG3* (Odgerel et al, 2010)), and four and a half LIM domain protein (*FHL1* (Selcen et al, 2011)) cause desmin-related (cardio-) myopathy (DRM) with intracellular protein accumulations. To date, DRM patients are treated symptomatically, and no causal treatment is available to halt this severe progressive disease involving myopathy, neuropathy, and cardiomyopathy. Different types of cardiomyopathy have been associated with DRM, such as dilated, hypertrophic, restrictive, and arrhythmogenic cardiomyopathies (Singh et al, 2020).

Independent of the genetic cause and type of cardiomyopathy, most DRM patients display intracellular accumulation of desmin and its chaperone αB-crystallin (CRYAB). Desmin is part of the type III intermediate filament protein group, interacting with myofibrils at the Z-disc, building a cellular scaffold that maintains a spatial relationship between the contractile apparatus and other components of the cardiomyocyte (Winter et al, 2014). Thus, its main functions include preserving cellular integrity, force transmission, and mechanochemical signaling (Su et al, 2022; Tsikitis et al, 2018). CRYAB is a small heat shock protein and chaperone that builds oligomeric complexes, interacts with desmin to support its proper folding, and prevents its aggregation (Claeyssen et al, 2024).

The *CRYAB* c.358 A > C (p.Arg120Gly; R120G) variant was first identified in a French family (Vicart et al, 1998) and has been intensively studied in rodent models (Bhuiyan et al, 2013; Sanbe et al, 2004; Wang et al, 2001; Xu et al, 2020). Mouse or rat cardiomyocytes overexpressing CRYAB[R120G] present large intracellular aggregates, disrupted cellular organization, and mitochondrial and proteolytic system impairment (McLendon and Robbins, 2011). CRYAB[R120G] mice develop cardiac hypertrophy and dilation, reduced ejection fraction with age, and die around 7–8 months (Bhuiyan et al, 2013; Wang et al, 2001). We previously conducted a high throughput screen to identify novel effectors that prevent CRYAB[R120G] protein aggregation in neonatal mouse cardiomyocytes (McLendon et al, 2017), and we identified Janus kinase 1 (JAK1) as a novel target in protein aggregation. JAK1 belongs to the family of non-receptor tyrosine kinases, along with JAK2, JAK3, and TYK2. The canonical function of JAKs is to transmit extracellular signals, such as cytokines binding to a receptor tyrosine kinase, via phosphorylation of Signal Transducers and Activators of

[1]Institute of Experimental Pharmacology and Toxicology, University Medical Center Hamburg-Eppendorf, Hamburg, Germany. [2]DZHK (German Centre for Cardiovascular Research), partner site North, Hamburg, Germany. [3]Technology Platform Microscopy and Image Analysis (TP MIA), Leibniz Institute of Virology (LIV), Hamburg, Germany. [4]Molecular Cardiovascular Biology, Cincinnati Children's Hospital and Medical Center, Cincinnati, USA. ✉E-mail: s.singh@uke.de

Transcription (STATs), which then translocate into the nucleus to regulate gene expression (Rawlings et al, 2004). The JAK-STAT pathway regulates cell proliferation, differentiation, and survival and is especially important for immune function and hematopoiesis (Rawlings et al, 2004). Thus, JAK variants can cause various diseases, from malignancies to autoimmune diseases (Luo et al, 2021). Therefore, several JAK inhibitors (Jakinibs) have recently been identified and tested in clinical trials with ruxolitinib (Jakafi®), the first FDA-approved Jakinib drug.

In this study, we investigated the effect of ruxolitinib and other Jakinibs on CRYAB$^{R120G}$ aggregation, the effect of ruxolitinib on the function of proteolytic systems, alterations in cardiomyocyte gene expression by JAK1 loss-of-function, and short-term ruxolitinib treatment or *Jak1* knockout in CRYAB$^{R120G}$ mice with DRM.

# Results

## Ruxolitinib treatment clears CRYAB$^{R120G}$ aggregates in neonatal rat ventricular myocytes (NRVMs)

RNA sequencing (RNAseq) of NRVMs and human induced pluripotent stem cell-derived cardiomyocytes (hiPSC-CMs) revealed that *Jak1/JAK1* and *Stat3/STAT3* are the most abundant of JAK-STAT family members (Table EV1), suggesting their important role in cardiomyocytes. Using different concentrations of the JAK inhibitor ruxolitinib in NRVMs, we showed that 3–10 µM ruxolitinib was sufficient to abolish phosphorylated STAT3 protein levels (Fig. EV1A). Since DRM patients already show aggregate accumulation in muscle biopsies when they seek health care, it is of great interest that a potential drug can prevent further protein aggregation at a late stage to blunt disease progression and ideally clear pre-existing aggregates to potentially heal from the disease. To visualize if ruxolitinib impacted CRYAB$^{R120G}$ aggregates, we performed live cell imaging in NRVMs transduced with CRYAB$^{R120G}$-GFP. Ruxolitinib (10 µM) was added at the beginning of the experiment, and the compound was not renewed due to technical limitations. Ruxolitinib treatment led to fewer CRYAB$^{R120G}$ aggregates after 2.5 days (endpoint; Fig. 1A–C). Strikingly, some of the ruxolitinib-treated cells exhibited aggregate dissolutions (Fig. 1A,B; marked with arrow; Movies EV1 and EV2). However, not all cells in the ruxolitinib-treated samples showed that dissolution. To analyze a higher number of cells for a more extended period, NRVMs were transduced with AdV5-CRYAB$^{R120G}$-GFP, kept in culture for 10 days, and fixed for analysis (Fig. EV1B,C). The aggregates of CRYAB$^{R120G}$ grew over time (Fig. EV1B,C). Ruxolitinib treatment from day 1 very efficiently prevented CRYAB$^{R120G}$ aggregate formation (EV1B,C). Adding ruxolitinib at later time points indicated that pre-existing aggregates could be cleared.

## JAK1 inhibitors result in fewer CRYAB$^{R120G}$ aggregates in human cardiomyocytes

To test whether ruxolitinib effectively reduces CRYAB$^{R120G}$ aggregates in human cardiomyocytes, we performed a live cell imaging experiment with hiPSC-CMs transduced with CRYAB$^{R120G}$-GFP. We observed that ruxolitinib also cleared pre-existing aggregates in some cells (Figs. 2A and EV2A; indicated by

arrows; Movies EV3 and EV4). Video analysis confirmed fewer CRYAB$^{R120G}$ aggregates in ruxolitinib-treated hiPSC-CMs at 36, 48, and 60 h (Fig. 2B). We then tested more selective JAK1 inhibitors—solcitinib, filgotinib (Jyseleca®), and upadacitinib (Rinvoq®)—in hiPSC-CMs. Filgotinib and upadacitinib are approved for rheumatoid arthritis and ulcerative colitis treatment. All three inhibitors are reversible ATP-binding site inhibitors with at least 5-fold higher selectivity for JAK1 compared to JAK2, JAK3, or TYK2. HiPSC-CMs were transduced with AdV5-CRYAB$^{R120G}$-GFP and treated with ruxolitinib, solcitinib, upadacitinib, filgotinib, or DMSO (vehicle control; Fig. 2C). All tested compounds lowered CRYAB$^{R120G}$ aggregate levels in hiPSC-CMs (Fig. 2C,D). Similar results were seen with solcitinib, upadacitinib, and filgotinib in NRVMs (Fig. EV2B). All inhibitors reduced CRYAB$^{R120G}$ aggregates by at least 50%. Therefore, JAK inhibition decreases CRYAB$^{R120G}$ aggregate load in human cardiomyocytes.

## Knockdown of *Jak1* and *Stat3* prevents CRYAB$^{R120G}$ aggregate formation

To evaluate whether the clearance of CRYAB protein aggregates is only induced by the JAK1 loss-of-function or if other pathway members are involved, we tested siRNAs targeting *Jak1*, or *Stat3* in NRVMs. Ruxolitinib treatment did not affect protein levels of JAK1 (Fig. 3A), JAK2 or STAT3, whereas it abolished STAT3 phosphorylation (Fig. EV1A). As expected, ruxolitinib treatment led to markedly lower aggregate load in IF images (Fig. 3A). Determination of protein levels in the soluble and insoluble fraction with or without ruxolitinib treatment revealed lower levels of desmin in the soluble fraction and lower levels of CRYAB$^{R120G}$-GFP and endogenous CRYAB in the insoluble fraction (Fig. 3B), supporting the IF findings. Similar results were obtained with the more selective JAK1 inhibitor upadacitinib (Fig. EV3E). JAK1 protein level was 60% lower in NRVMs treated with siJak1 than with scramble siRNA (scr; Fig. 3C). This was associated with markedly lower CRYAB$^{R120G}$ aggregate load, suggesting that a partial reduction of JAK1 function is sufficient to prevent aggregate formation. The siJak1 treatment did not affect *Jak2* expression levels and vice versa (Fig. EV3A). *Jak1* knockdown or ruxolitinib treatment did not result in lower GFP levels in a control experiment (Fig. EV3B–D). As expected, siJak1 treatment resulted in lower CRYAB$^{R120G}$-GFP levels determined by Western blot (Fig. EV2D,E). STAT3 protein level and CRYAB$^{R120G}$ aggregate load were both markedly lower in NRVMs treated with siStat3 than with scr (Fig. 3D), suggesting that reduction of CRYAB$^{R120G}$ aggregates is mediated through JAK1-STAT3.

## UPS-mediated degradation is enhanced by ruxolitinib treatment, *Jak1* or *Stat3* knockdown

Since proteolytic pathways such as the UPS or autophagy-lysosomal pathway (ALP) play a significant role in the removal of protein accumulations and aggregates, we tested the effect of ruxolitinib, siJak1 or siStat3 treatment on UPS- or ALP-mediated degradation. To our knowledge, the regulation of UPS- or ALP-mediated degradation by ruxolitinib has not yet been reported. To assess the UPS-mediated degradation, NRVMs were transduced with an AdV5-GFPu (inverse UPS reporter), transfected with siRNA targeting proteasome 26S subunit, non-ATPase 1 (*Psmd1*;

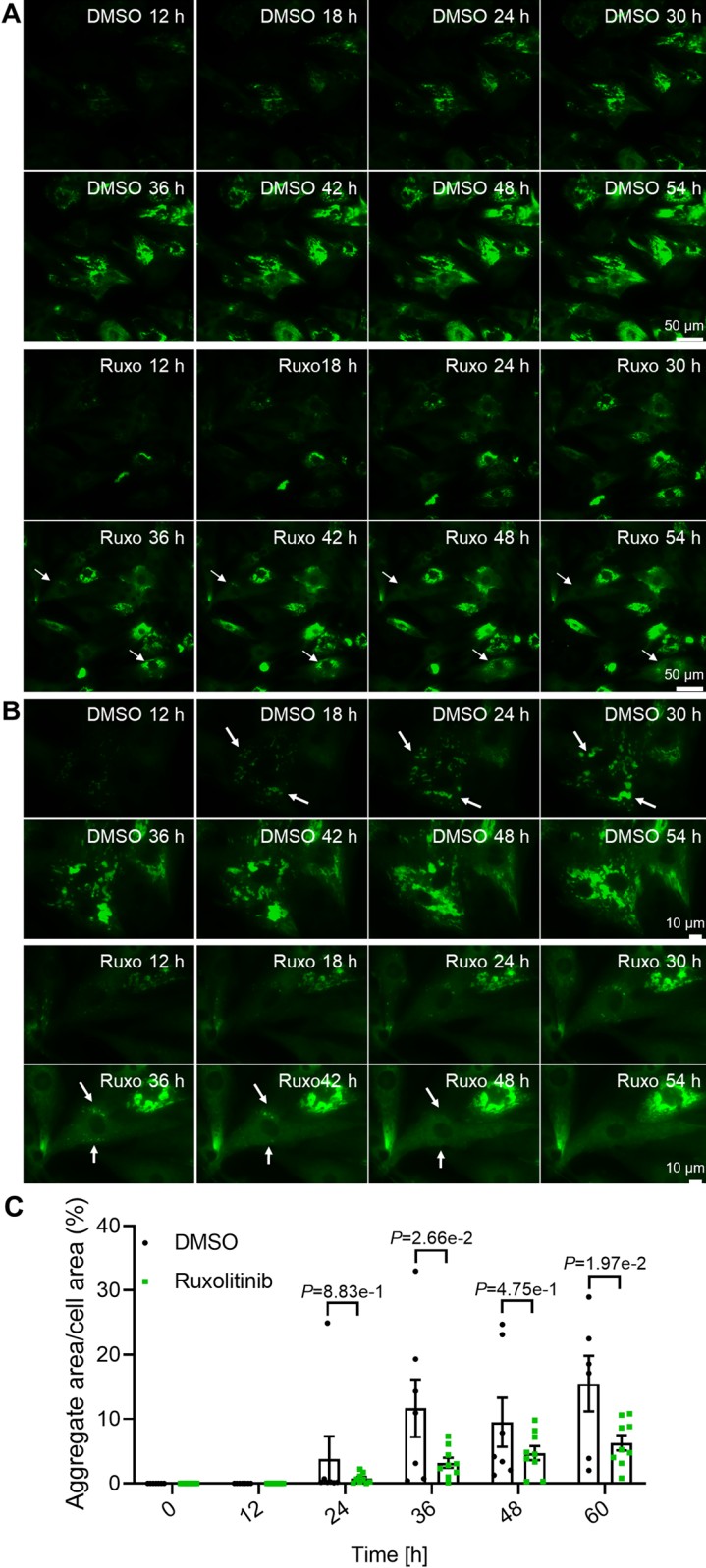

◀ **Figure 1. Ruxolitinib dissolves CRYAB^{R120G} aggregates in NRVMs.**

Live cell imaging of NRVMs transduced with AdV5-CMV-CRYAB^{R120G}-GFP and treated with 10 μM ruxolitinib (ruxo) or DMSO. (A) Representative fluorescence images. Scale bar = 50 μm. (B) Zoom of representative cardiomyocytes. Aggregates are depicted in green (CRYAB^{R120G}-GFP). Arrows mark growing or dissolving aggregates. Scale bar = 10 μm. (C) Quantification of aggregates in F-actin area (SiR-actin staining) with NIS Elements software. Each data point represents a whole image with a number of cells. Data were obtained from 8–10 points from 2 wells per condition and 1 NRVM preparation and are depicted as mean ± SEM, and *p*-values were obtained with two-way ANOVA and Sidak's multiple comparisons post-hoc analysis. Corresponding videos can be found in the online supplements (Movies EV1 and EV2). Source data are available online for this figure.

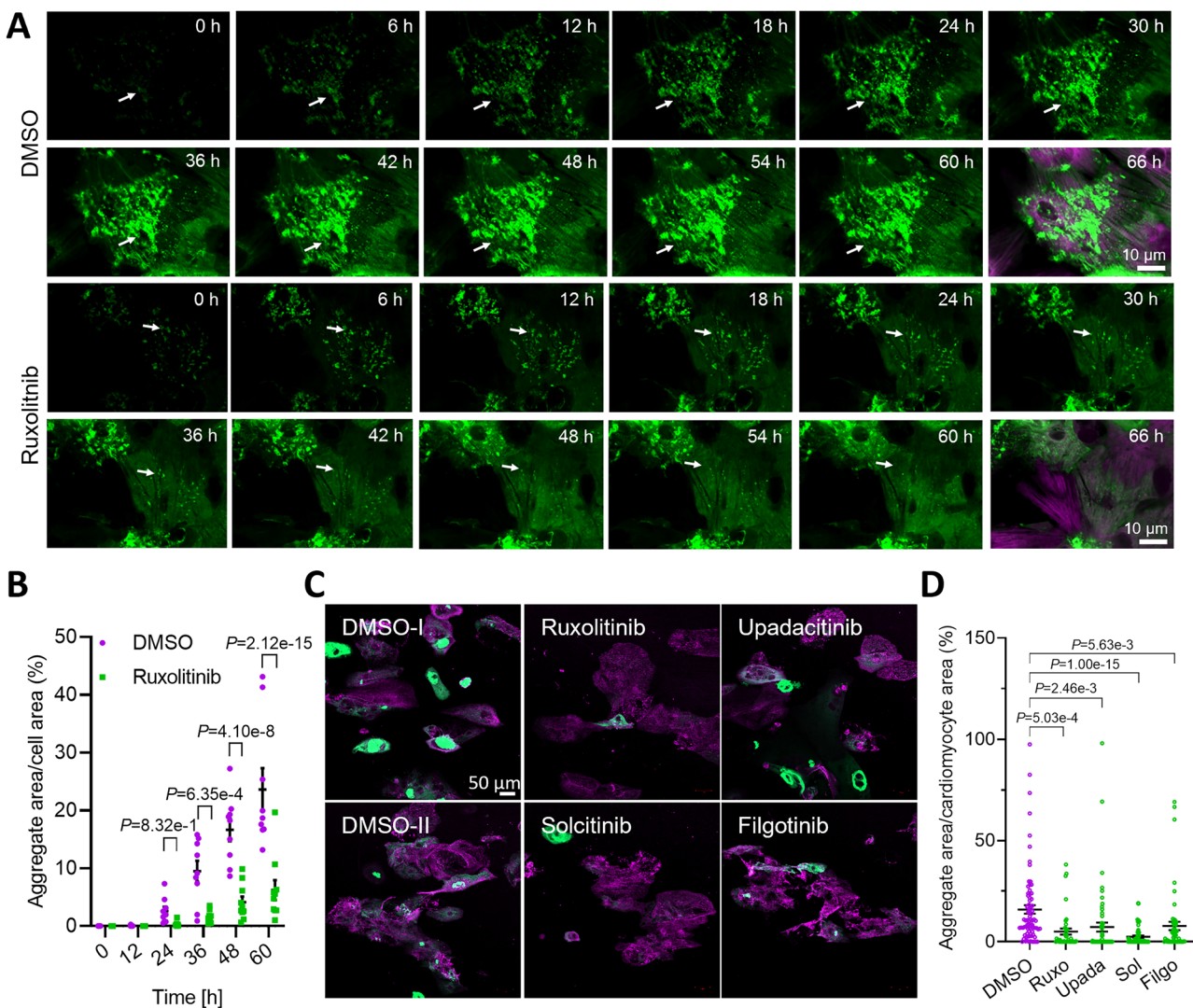

**Figure 2. JAK1 inhibitors result in fewer CRYAB^{R120G} aggregates in human cardiomyocytes.**

(A) HiPSC-CMs transduced with AdV5-CMV-CRYAB^{R120G}-GFP were treated with 10 μM ruxolitinib or 0.1% DMSO at the start of the long-term time-lapse live cell imaging (confocal mode). Representative images. Scale bar = 10 μm. Aggregates are depicted in green (CRYAB^{R120G}-GFP) and cells in purple (SiR-actin). Arrows indicate moving and forming, or dissolving aggregates. The corresponding videos can be found in the supplements (Movies EV3 and EV4). (B) Quantification of aggregates in F-actin-positive cells of videos from A with NIS Elements software. Data were obtained from one hiPSC-CM differentiation with two wells per condition and 5–8 selected points per well and are depicted as mean ± SEM, with *p*-values from two-way ANOVA and Sidak's post-hoc analysis. (C) Representative IF images of fixed hiPSC-CMs transduced with AdV5-CMV-CRYAB^{R120G}-GFP and treated with 3 μM ruxolitinib (Ruxo), 1 μM upadacitinib (Upada), 1 μM solcitinib (Sol), 1 μM filgotinib (Filgo) or 0.1% DMSO every other day for 1 week. Aggregates are depicted in green (CRYAB^{R120G}-GFP) and cardiomyocytes in purple (anti-ACTN2). Scale bar = 50 μm. (D) Quantification of aggregates in cardiomyocytes (hiPSC-CMs) from (B) with ImageJ software. Data were obtained from 2 independent hiPSC-CM differentiations with at least 4 wells per condition and 5–8 images per well, and are depicted as mean ± SEM, with *p*-values obtained by the one-way ANOVA and Dunnett's post-hoc analysis. Source data are available online for this figure.

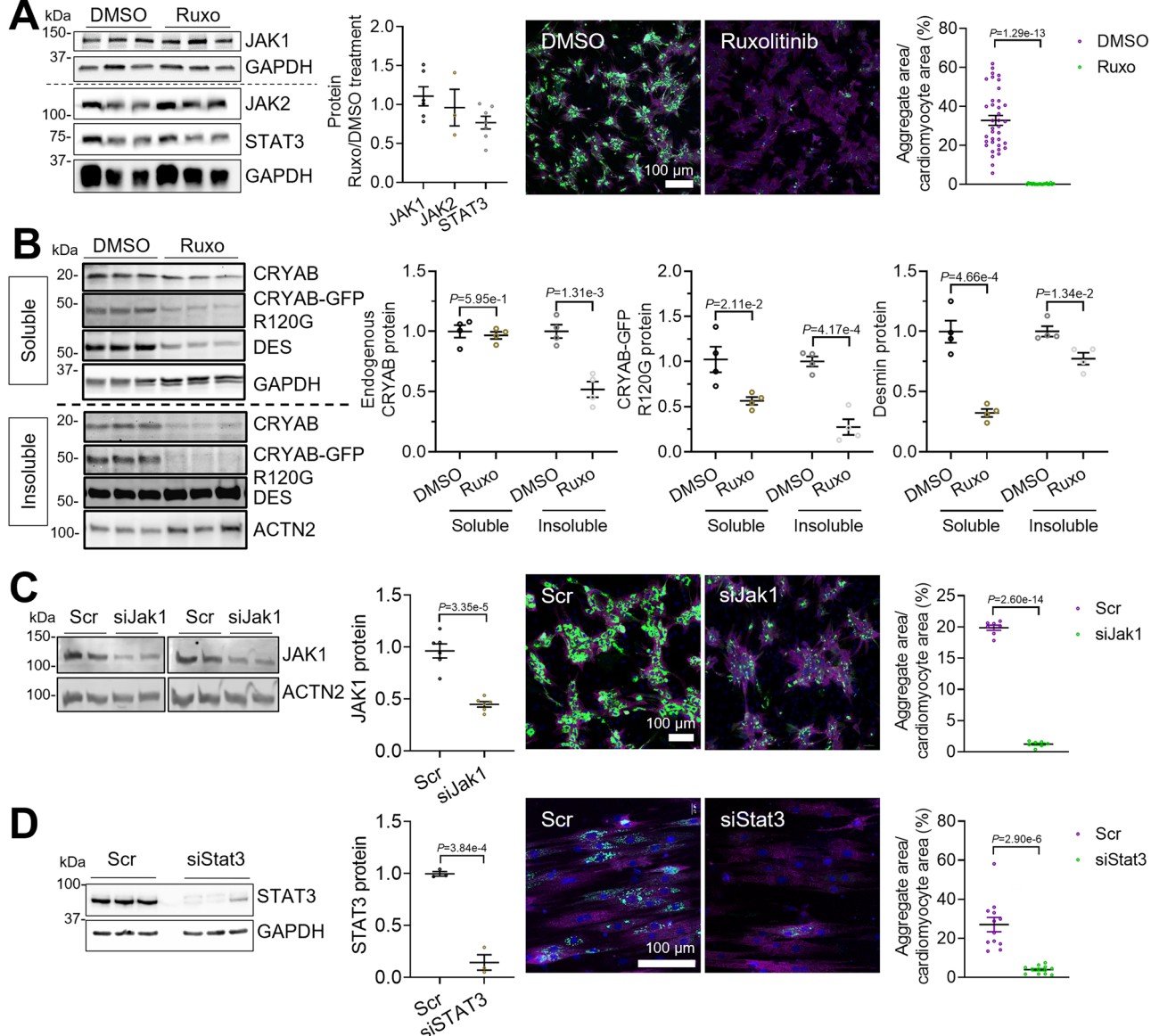

**Figure 3. Knockdown of *Jak1* or *Stat3* and treatment with ruxolitinib results in markedly lower CRYAB$^{R120G}$ aggregate load.**

NRVMs were transduced with AdV5-CMV-CRYAB$^{R120G}$-GFP and treated with 3 μM ruxolitinib or DMSO or 100 nM siRNAs targeting either *Jak1* (siJak1) or *Stat3* (siStat3), or scramble siRNA (scr). Thereafter, NRVMs were harvested or fixed after 4–6 days. (A, B) Treatment with ruxolitinib or DMSO. (C) Transfection with siJak1. (D) Transfection with siStat3. (A–D) Western blots of protein extracts from treated NRVMs were stained with antibodies directed against indicated proteins. In the representative immunofluorescence images, aggregates are depicted in green (CRYAB$^{R120G}$-GFP), cardiomyocytes in purple (anti-cardiac troponin I), and nuclei in blue (DAPI). Scale bar = 100 μm. Quantification of aggregates in cardiomyocytes with NIS Elements or ImageJ software. Data were obtained with a minimum of 3 replicates per condition. The number of replicates is indicated by the dots. Data are depicted as mean ± SEM, and *p*-values were obtained with the unpaired Student's *t*-test. Source data are available online for this figure.

knockdown of UPS function) or scr, and treated with ruxolitinib or DMSO (Fig. 4A). As expected (Singh et al, 2021), the efficient knockdown of *Psmd1*, encoding the essential 26S proteasome non-ATPase regulatory subunit 1 (PSMD1), led to higher GFPu protein levels (Fig. 4A), suggesting an impaired UPS-mediated degradation. In line with previous findings (Chen et al, 2005), transduction of NRVMs with AdV5-CRYAB$^{R120G}$-GFP also impaired UPS-mediated degradation (Fig. 4A). Ruxolitinib treatment resulted in markedly lower GFPu protein levels in CRYAB$^{R120G}$-GFP-transduced cells,

suggesting that it positively regulates UPS-mediated degradation (Fig. 4A). This effect of ruxolitinib was abolished when NRVMs were treated with siPsmd1 (Fig. 4A). In line with these results, siJak1 treatment resulted in a markedly lower GFPu protein level, which was blunted with *Psmd1* knockdown (Fig. 4B). Similarly, *Stat3* knockdown led to lower GFPu protein levels (Fig. 4C). To test whether the effect of ruxolitinib is mediated through its inhibition of JAK1, the effect of ruxolitinib on GFPu in NRVMs silenced for JAK1 was assessed (Fig. 4D). Ruxolitinib led to lower GFPu levels

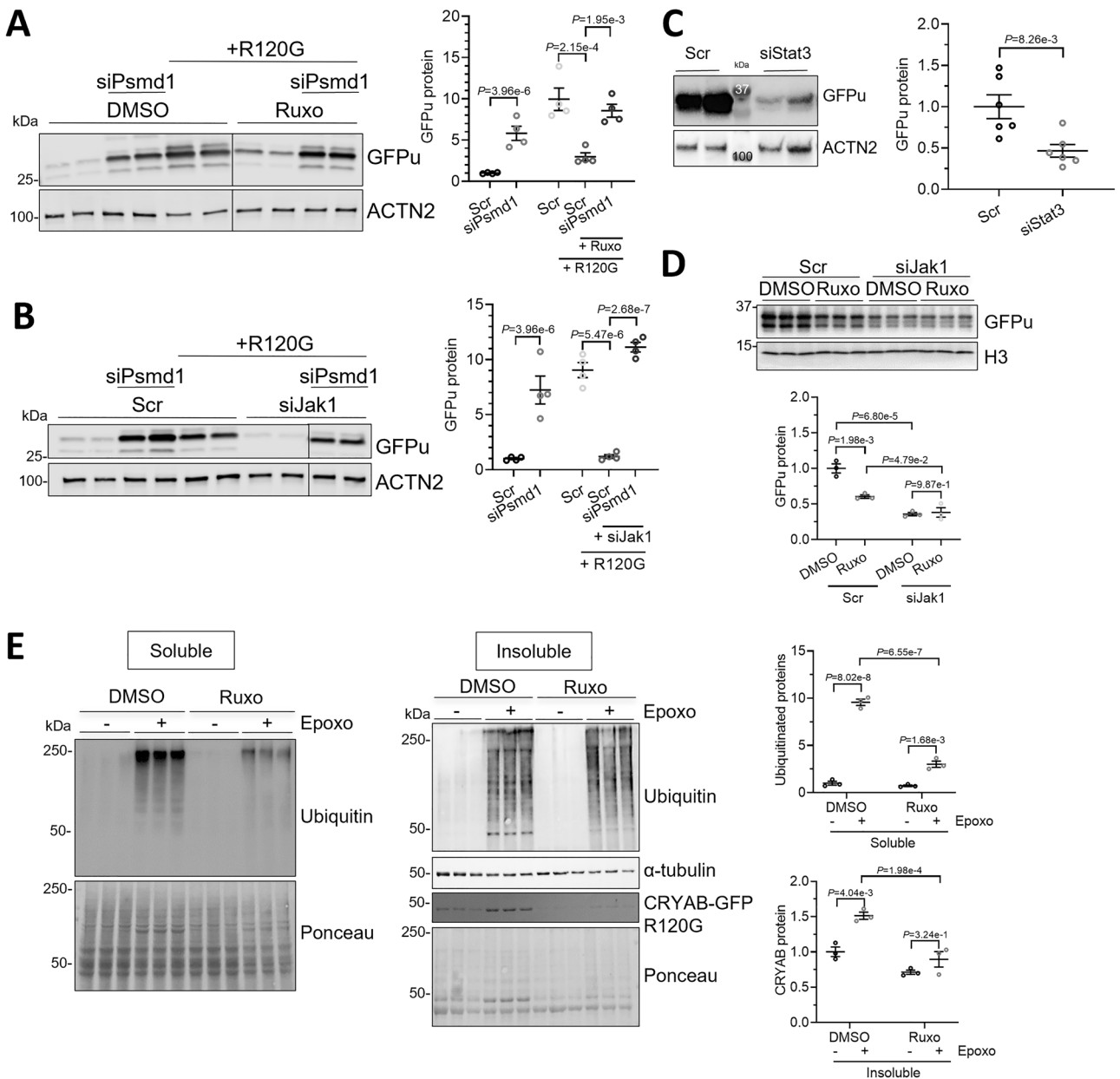

**Figure 4. UPS-mediated degradation is enhanced by ruxolitinib treatment, *Jak1* or *Stat3* knockdown.**

NRVMs were transduced with AdV5-CMV-GFPu for 5 days and/or AdV5-CMV-CRYAB^R120G where indicated. (A) NRVMs transfected with 20 nM siPsmd1 or scramble siRNA (scr) and treated with 3 μM ruxolitinib (ruxo) or DMSO for 5 days (every other day). (B) NRVMs transfected with 20 nM siPsmd1 or scr and 100 nM siJak1 or scr. (C) NRVMs transfected with 100 nM siStat3 or scr. (D) NRVMs were transfected with 100 nM siJak1 or scr and/or treated with 3 μM ruxolitinib (ruxo). (E) NRVMs were treated with 3 μM ruxolitinib (ruxo) or DMSO. Cells were treated with epoxomicin for 15 h and were harvested and fractionated into soluble and insoluble fraction. (A–E) WBs were stained with antibodies against indicated proteins of protein extracts from treated NRVMs. WB quantification with Image Studio™, Image Lab™ or Bio-1D-Vilber software. Data were obtained with a minimum of 3 replicates per condition. The number of replicates is indicated by the dots. Data are depicted as mean ± SEM, and *p*-values were obtained with the one or two-way ANOVA and Tukey's multiple comparisons post-hoc or with the unpaired Student's t-test (C). Source data are available online for this figure.

in scramble siRNA-treated cells as expected. Ruxolitinib did not lead to any additional lowering of GFPu levels in NRVMs treated with siJak1, suggesting that JAK1 loss-of-function is responsible for the enhanced UPS function after ruxolitinib treatment. To evaluate the effect of ruxolitinib independent of GFPu, we included the total level of ubiquitinated proteins in the soluble and insoluble fraction

of Ruxo- or DMSO-treated NRVMs (Fig. 4E). The ubiquitinated protein levels were markedly lower with ruxolitinib and epoxomicin treatment compared to the control in the soluble fraction. This result indicates that ruxolitinib treatment enhanced UPS activity, thus leading to fewer ubiquitinated proteins accumulating in the cells. In summary, these experiments suggest a positive regulation

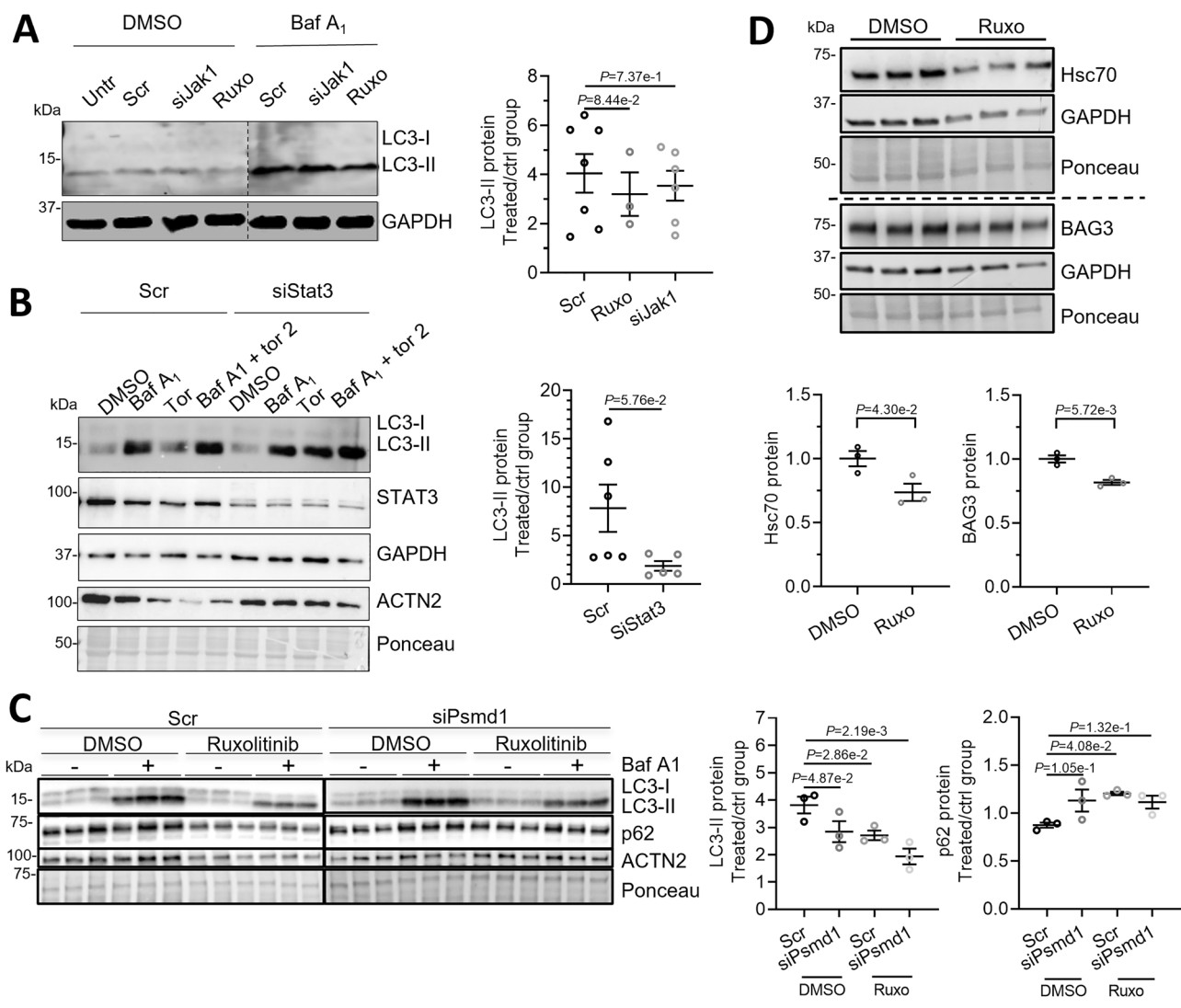

**Figure 5. No major effect on autophagic activity by ruxolitinib treatment, *Jak1* or *Stat3* knockdown.**

NRVMs were transduced with AdV5-CMV-GFPu for 5 days and/or AdV5-CMV-CRYAB[R120G] where indicated. (A) NRVMs transfected with 20 nM siJak1 or scramble siRNA (scr) and treated with 3 μM ruxolitinib (ruxo) or DMSO for 5 days (every other day). (B) NRVMs transfected with 100 nM siStat3 or scr and 100 nM siJak1 or scr. (C) NRVMs were transfected with 20 nM siPsmd1 and treated with 3 μM ruxolitinib. (D) NRVMs were treated with 3 μM ruxolitinib (ruxo) or DMSO. (A–D) WBs were stained with antibodies against the indicated proteins of protein extracts from treated NRVMs. WB quantification with Image Studio™ or Image Lab™ or Bio-1D-Vilber software. Data were obtained with a minimum of 3 replicates per condition. The number of replicates is indicated by the dots. Data are depicted as mean ± SEM, and p-values were obtained with the two-way ANOVA and Tukey's multiple comparisons post-hoc or with the unpaired Student's t-test (D). Source data are available online for this figure.

of UPS-mediated degradation by ruxolitinib, mediated through a loss of JAK1 and STAT3 function.

## Autophagic activity is not responsible for aggregate clearance after ruxolitinib treatment, *Jak1* or *Stat3* knockdown

To assess ALP-mediated degradation, NRVMs were treated with the lysosomal V-ATPase inhibitor bafilomycin A1 or DMSO to assess the autophagic flux via the quantification of the autophagy marker microtubule-associated protein 1 light chain 3 (LC3-II) by Western blot analysis. As expected, bafilomycin A1 treatment resulted in a marked accumulation of LC3-II and a fold change of

~4 for the autophagic flux (Fig. 5A). Ruxolitinib treatment did not affect the autophagic flux (Fig. 5A). In line with our previous study (McLendon et al, 2017), *Jak1* knockdown also did not affect the autophagic flux (Fig. 5A). This experiment was repeated with *Stat3* knockdown and combining bafilomycin A$_1$ and torin 2, an autophagy activator to exaggerate the autophagic flux (Fig. 5B). With *Stat3* knockdown, there was no significant effect on the autophagic flux. We performed an additional experiment with knockdown of the essential proteasomal subunit PSMD1 (siPsmd1) in combination with ruxolitinib treatment (Fig. 5C) to test if autophagy is activated with ruxolitinib treatment when the UPS is inhibited. However, we observed a rather inhibited autophagic flux after siPsmd1 and ruxolitinib treatment (Fig. 5C), further

indicating that ruxolitinib treatment does not activate autophagy. Furthermore, we evaluated levels of proteins involved in another form of autophagy, chaperone-mediated autophagy, by quantifying protein levels of Heat-shock cognate 70 kDa protein (Hsc70) and Bcl-2-associated athanogene 3 (BAG3) by WB (Fig. 5D). Hsc70 and BAG3 protein levels were slightly lower in ruxolitinib- than DMSO-treated cells. Overall, all results related to autophagy indicate a mildly reduced autophagic activity in cells treated with ruxolitinib. These results suggest that CRYAB$^{R120G}$ aggregate removal with ruxolitinib treatment is independent of autophagic activity but depends on UPS activity.

## The E3 ligase ASB2β is responsible for CRYAB$^{R120G}$ aggregate reduction with ruxolitinib treatment

To evaluate if the enhanced UPS-mediated degradation is responsible for the lower CRYAB$^{R120G}$ aggregate load, NRVMs were transduced with AdV5-CRYAB$^{R120G}$-GFP, transfected with scramble RNA or siPsmd1 to block UPS-mediated degradation, and treated with ruxolitinib or *Jak1* knockdown (Fig. 6A). In DMSO condition, CRYAB$^{R120G}$ aggregate did not differ between *siPsmd1*- or scramble siRNA-treated NRVMs (Fig. 6A), probably due to the high aggregate load and already maximally impaired UPS. In contrast, in ruxolitinib-treated NRVMs, *Psmd1* knockdown resulted in CRYAB$^{R120G}$ aggregate accumulation (Fig. 6A), indicating that the ruxolitinib-induced enhancement of UPS-mediated degradation is responsible for the removal of the CRYAB$^{R120G}$ aggregates. The experiment was repeated with *Jak1* knockdown with similar results (Fig. 6A), supporting the findings.

Since the JAK-STAT pathway regulates gene expression, paired-end RNAseq was performed on RNA extracts from NRVMs transfected with siRNA directed against *Jak1* or scramble siRNA. Data were analyzed using a differential gene expression analysis method. As a result, 117 RNAs were upregulated and 92 downregulated after *Jak1* knockdown (Fig. EV4A). As expected with modulation of the JAK-STAT pathway, expression of genes involved in biological processes such as cell differentiation, cytokine production, and inflammatory response (Fig. EV4B), molecular functions such as phosphotransferase activity, chemokine activity, and receptor binding (Fig. EV4C), and pathways such as interferon-gamma and interleukin signaling (Fig. EV4D) were primarily affected. When mapping the identified RNAs to data sets (Singh et al, 2021) with gene identifiers of proteins involved in the UPS or ALP, a match with 5 ubiquitinating enzymes was found (Fig. EV4E). In addition, RNAseq results were analyzed for important muscle E3 ubiquitin ligases (Peris-Moreno et al, 2021) (Table EV2). We found significantly higher mRNA levels for ankyrin repeat and SOCS box-containing 2 (*Asb2*), F-box protein 40 (*Fbxo40*), membrane-associated ring-CH-type finger 3 (*March3*), receptor-associated protein of the synapse (*Rapsn*), ring finger protein 152 (*Rnf152*), ring finger protein 207 (*Rnf207*), tripartite motif containing 50 (*Trim50*) and a tendency to higher RNA levels for F-box and leucine-rich repeat 21 (*Fbxl21*), and muscle-specific ring finger protein 2 (*Murf2/Trim55*), suggesting UPS activation through upregulating gene expression of E3 ubiquitinating ligases. An introduction into the function of these E3 ligases and their potential interaction with CRYAB is given in Table EV2.

We performed siRNA knockdown experiments of *Asb2*, *Rapsn*, *Trim50*, and *Rnf207* in combination with siJak1 in NRVMs. *Rnf152*, *March3*, and *Fbxo40* were not tested due to low expression levels, downregulation by siJak1 or ruxolitinib or failed siRNA knockdown, respectively. Whereas *Raspn*, *Trim50*, and *Rnf207* knockdown did not reveal any major difference in aggregate load after siJak1 treatment (Fig. EV5), *Asb2* knockdown abolished the effect of ruxolitinib on aggregate reduction (Fig. 6A). It has been shown previously that ASB2β targets desmin and filamin C for proteasomal degradation (Thottakara et al, 2015), both proteins, which were shown to accumulate together with CRYAB$^{R120G}$ into aggregates (Feldkirchner et al, 2012). We found that *Asb2* knockdown also resulted in higher endogenous CRYAB levels in untransduced NRVMs (Fig. 6B), prompting the question of whether ASB2 also targets CRYAB for degradation. We performed experiments with wild-type (WT) and mutant (L595A) ASB2β. The mutant form lacks the ligase activity and thus impairs the degradation of ASB2β substrates. We validated in our model that desmin accumulates if mutant ASB2β is overexpressed (Fig. 6C). Furthermore, the overexpression of WT ASB2β in combination with *Psmd1* knockdown induced CRYAB accumulation in the soluble fraction (Fig. 6D), indicating that CRYAB is degraded by the proteasome. Conversely, CRYAB level was about 3-fold higher with L595A than WT ASB2β in any condition and did not accumulate further with L595A ASB2β and *Psmd1* knockdown (Fig. 6D), indicating that the ligase function of ASB2β is necessary for the proteasomal degradation of CRYAB. In addition, when we pulled down WT or L595A ASB2β, we found that CRYAB co-immunoprecipitated (Fig. 6E), suggesting that it binds to ASB2β. Moreover, when we performed a ubiquitin pulldown and stained for CRYAB, we observed lower levels of ubiquitinated CRYAB with L595A ASB2β than with WT ASB2β (Fig. 6F). In summary, our results suggest that JAK1 loss-of-function results in higher expression of ASB2β, which then increasingly targets desmin and CRYAB for proteasomal degradation, resulting in fewer aggregates.

## Higher phosphorylated STAT3 protein levels in DRM mice with hypertrophy and dysfunction

To test if JAK-STAT signaling is pathologically altered in CRYAB$^{R120G}$ DRM mice, we analyzed the hearts of DRM mice and compared them to their non-transgenic (NTG) littermates. DRM mice did not exhibit cardiac hypertrophy at 1 month, whereas hypertrophy was fully established at 4 months and did not increase further with age (Fig. 7A). Left ventricular ejection fraction (LVEF) was lower in DRM mice at 7 months of age (Fig. 7B). Therefore, we determined phosphorylated STAT3 (P-STAT3) protein levels in 1-, 4- and 7-month-old mice. The P-STAT3/STAT3 ratio did not differ in 1-month-old but was 2- and 5-fold higher in 4- and 7-month-old DRM than NTG mice, respectively (Fig. 7C,D), suggesting pathological activation of the pathway with age. *Stat3* mRNA levels did not differ between the groups at 7 months (Fig. 7E). In addition, we performed soluble and insoluble fractionation of proteins extracts from 7-month-old CRYAB R120G and NTG mice, and found the levels of CRYAB, desmin, and P-STAT3 to be higher in the soluble and insoluble fraction, and the levels of JAK1 to be higher in the insoluble fraction of R120G mice (Fig. 7F).

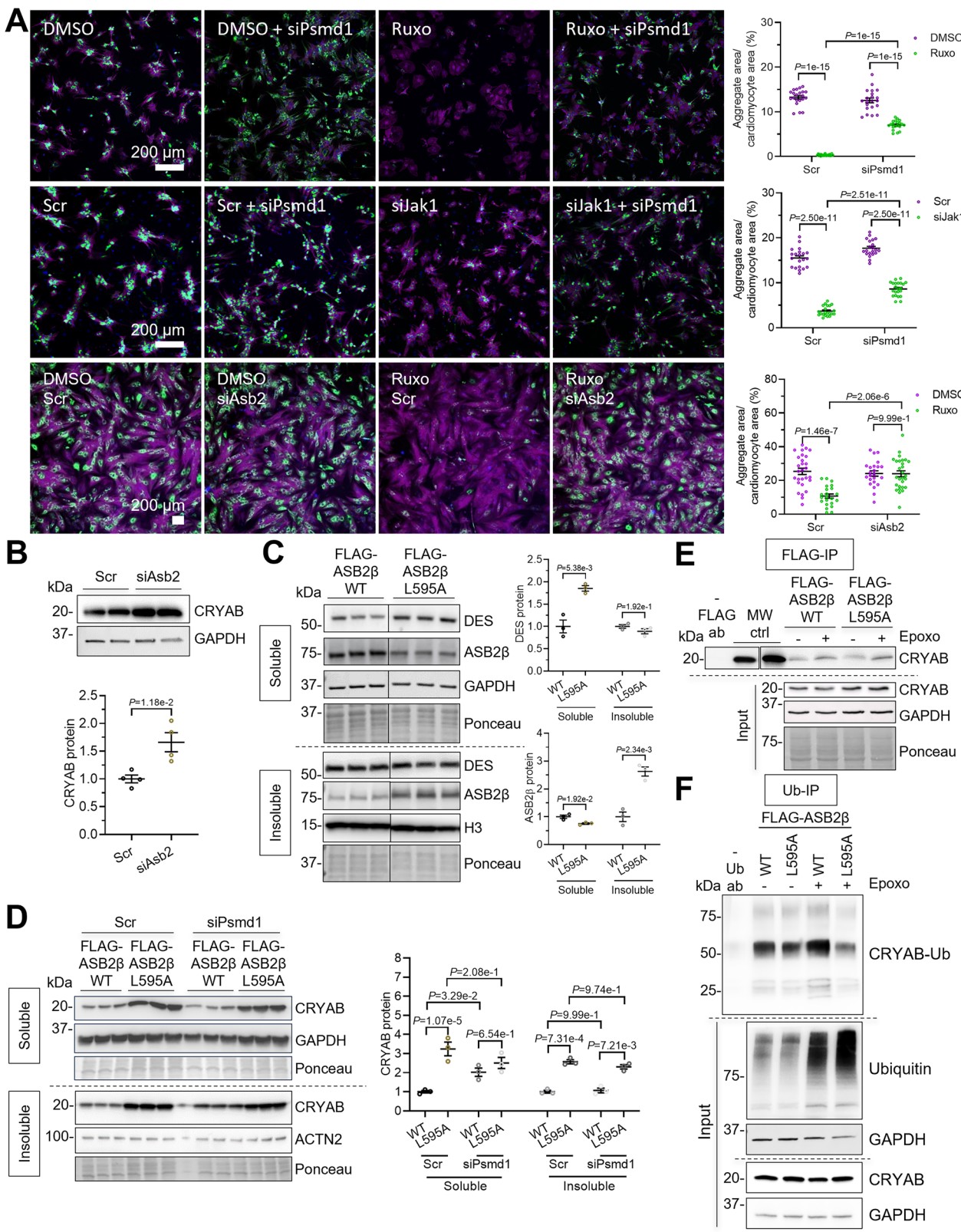

**Figure 6. The E3 ligase ASB2β is responsible for CRYAB^R120G aggregate reduction with ruxolitinib treatment.**

(A) NRVMs transfected with siRNAs targeting essential proteasome 26S subunit, non-ATPase 1 (siPsmd1; 20 nM), siAsb2 (50 nM) or scramble siRNA (scr), transduced with AdV5-CMV-CRYAB^R120G, treated with 3 µM ruxolitinib (ruxo) or DMSO, or transfected with 100 nM siJak1 or scr and fixed after 6 days. Representative immunofluorescence images; scale bar = 200 µm. Aggregates are depicted in green (CRYAB^R120G-GFP), cells in purple (anti-cardiac troponin I or phalloidin), and nuclei in blue (DAPI). Quantification of aggregates in cardiomyocytes with NIS Elements software. Dots represent the number of analyzed images. (B) Western blot of protein extracts of NRVMs transfected with siAsb2 (50 nM) or scramble siRNA (scr) for 5 days. (C–F) NRVMs were transduced with plasmids overexpressing the wild type (WT) FLAG-ASB2β or the inactive (L595A) FLAG-L595A-ASB2β and harvested after 5 days for Western blot analysis or co-immunoprecipitation (co-IP). (C, D) Western blots and quantifications of NRVMs after fractionation into soluble and insoluble. (E) Western blot of FLAG-IP stained with anti-CRYAB, and corresponding input samples. (F) Western blot of Ub-IP stained with anti-CRYAB, and corresponding input samples. WB quantification with Image Studio™ or Image Lab™, or Bio-1D-Vilber software. Data were obtained with a minimum of 3 replicates per condition. The number of replicates is indicated by the dots. Data are depicted as mean ± SEM, and p-values were obtained with the two-way ANOVA and Tukey's multiple comparisons post-hoc analysis or with the unpaired Student's t-test (B). Source data are available online for this figure.

## Evaluation of potential cardiotoxic effects of ruxolitinib in human engineered heart tissues (EHTs)

The long-term adverse effects of ruxolitinib over several years have been monitored across multiple clinical/observational studies (Aruga et al, 2025; Kiladjian et al, 2020; Verstovsek et al, 2017), with the most common effects including anemia, low platelet count, and increased risk of infection. However, potential cardiotoxicity has not been well-documented. Direct cardiotoxic effects, such as myocardial injury or QT prolongation, were not reported, and major adverse cardiovascular reactions with ruxolitinib treatment have been rarely reported in clinical studies. To address this gap, we assessed the effects of ruxolitinib on cardiac contractility using control hiPSC-derived engineered heart tissues (hiPSC-EHTs) (Fig. 8). EHTs represent a well-established, FDA-accepted platform for preclinical cardiotoxicity testing. Reported ruxolitinib plasma concentrations in patients range from 0.2–1.7 µM, depending on the dosing (Appeldoorn et al, 2023). We treated EHTs with increasing concentrations of ruxolitinib (0, 0.3, 1, 3, 10, and 30 µM) for four weeks, starting 30 days after casting, to model chronic exposure to regular and high ruxolitinib concentrations. As expected, ruxolitinib reduced STAT3 signaling, with P-STAT3 levels markedly decreased at 0.3 µM and undetectable at ≥1 µM (Fig. 8A). At treatment initiation, EHT force generation had plateaued. Chronic exposure to 0.3–3 µM ruxolitinib did not alter contractile force or spontaneous beating rate over time (Fig. 8B), indicating no detectable toxicity within this concentration range. By contrast, 30 µM ruxolitinib led to progressive reductions in force and beating frequency, consistent with cardiotoxic effects. Treatment with 10 µM ruxolitinib produced a non-significant trend toward decreased force and beating rate compared with DMSO controls (Fig. 8B). Contractile kinetics were assessed under electrical pacing at day 40 and day 60 (Fig. 8C,D). There were no significant differences in absolute force among all treatment groups, although there was a clear trend towards lower force with 30 µM ruxolitinib. In addition, 30 µM ruxolitinib slightly shortened contraction time (time-to-peak 80%) and prolonged relaxation time (relaxation 80%). Histological analysis at the study endpoint revealed no major structural differences among groups (Fig. 8E). In summary, our data indicate that ruxolitinib is well tolerated by EHTs at concentrations up to 10 µM, with evidence of cardiotoxicity emerging only at ≥30 µM levels, approximately 30-fold higher than those measured clinically.

## Ruxolitinib treatment prevents the development of cardiac dysfunction in CRYAB^R120G transgenic mice

To test if ruxolitinib blunts cardiac disease in DRM mice, 21-week-old DRM and NTG mice were treated twice daily with 75 mg/kg ruxolitinib or vehicle by oral gavage for 3 weeks with initial and final echo (Fig. 9A). P-STAT3/STAT3 ratio was lowered by ruxolitinib treatment (Fig. 9B). At the beginning of the experiment, cardiac function, represented by LVEF, did not differ between DRM and NTG mice (Fig. 9C). As anticipated, LVEF was lower in vehicle-treated DRM than NTG mice at the end of the experiment. Ruxolitinib treatment prevented the drop in LVEF in DRM mice (Fig. 9C). Ruxolitinib treatment prevented an increase in left ventricular end-systolic volume (LVESV) in R120G mice, indicating better cardiac function. Left ventricular internal diameter in systole (LVIDs), left ventricular end-diastolic volume (LVEDV), left ventricular internal diameter in diastole (LVIDd), and left ventricular mass-to-body weight (LVM/BW) were not affected by ruxolitinib treatment. Similar to LVM/BW, cardiac hypertrophy represented by heart weight (HW) and heart weight-to-body weight ratio (HW/BW) were higher in DRM than in NTG mice and were not affected by ruxolitinib treatment (Fig. 9D). Body weights did not differ between the groups (Fig. 9D). CRYAB^R120G aggregate load was lower in ruxolitinib than in control-treated mice, determined by immunofluorescence (Fig. 9E,F) and Western blot (Fig. 9G). Overall desmin structure was improved by ruxolitinib treatment (Fig. 9H, indicated by green arrows).

## Conditional cardiomyocyte-specific *Jak1* knockout prevents the development of cardiac dysfunction in CRYAB^R120G transgenic mice

To test if cardiomyocyte-specific *Jak1* knockout (KO) blunts cardiac disease in DRM mice, 8-week-old DRM and NTG mice crossed with conditional *Jak1* KO and cardiomyocyte-specific αMHC-MerCreMer mice were fed with tamoxifen chow to induce KO and analyzed by echocardiography and heart extraction at 26 weeks of age (Fig. 10A). *Jak1* mRNA levels were about 40% and 70% lower in induced heterozygous and homozygous cardiomyocyte-specific *Jak1* KO mice, respectively (Fig. 10B). JAK1 protein levels were about 70% lower in induced homozygous *Jak1* KO mice (Fig. 10C). It should be noted that *Jak1* KO is only induced in cardiomyocytes, and other cells of the heart still express *Jak1*. As anticipated, LVEF was lower in 26-week-old DRM than NTG mice with *Jak1* wt (Fig. 10D). Heterozygous *Jak1* KO did not have a major effect on LVEF in NTG or DRM mice (Fig. 10D). Strikingly, in the homozygous *Jak1* KO group, LVEF did not differ between DRM and NTG mice (Fig. 10D). LVEF did not differ between *Jak1* KO and wt in the NTG group. Like ruxolitinib treatment in mice, homozygous *Jak1* KO blunted an increase in LVESV in R120G mice, suggesting preservation of cardiac function (Fig. 10D). Like in ruxolitinib-treated mice, LVIDs, LVEDV,

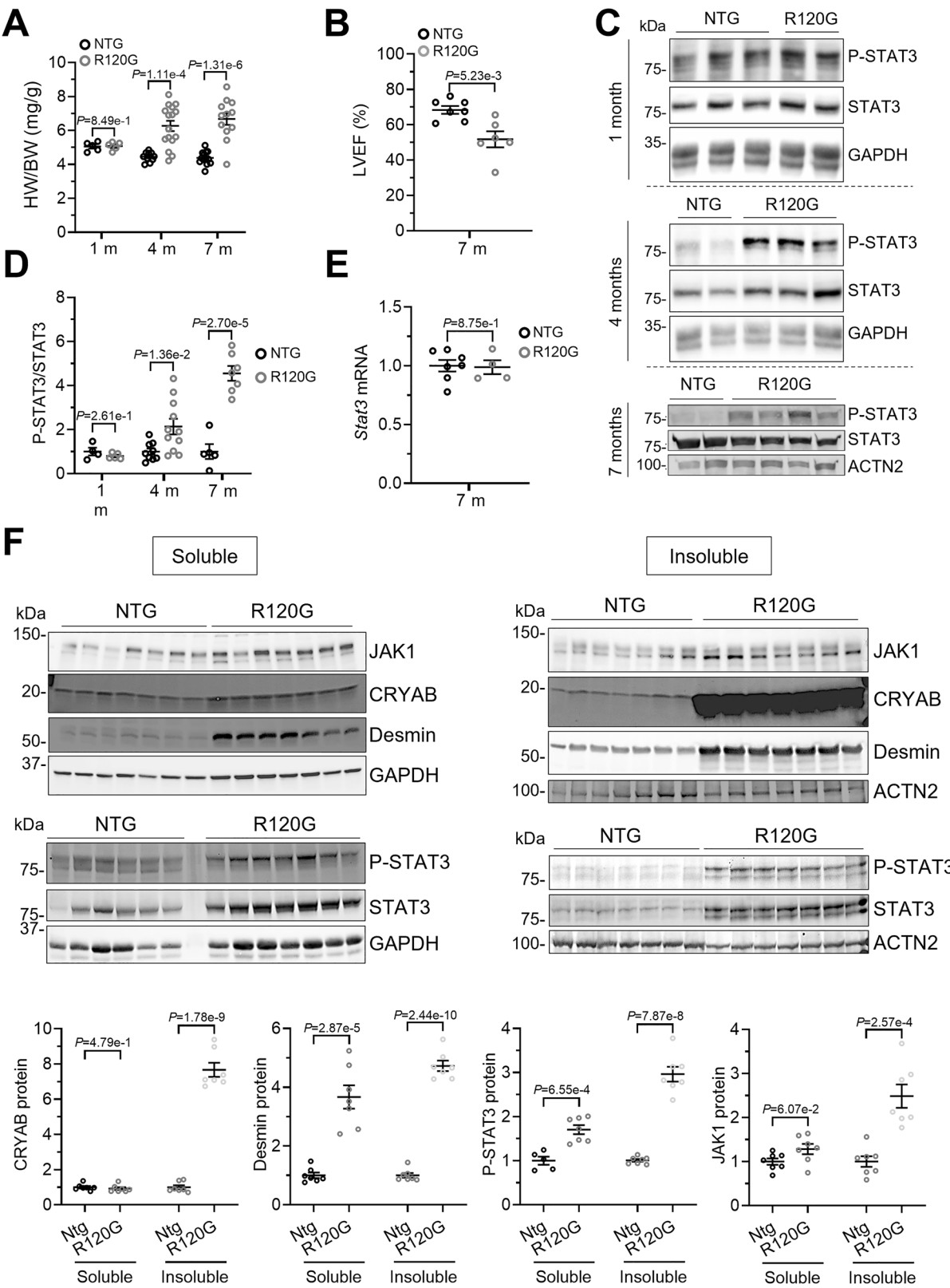

 **Figure 7. Phosphorylated STAT3 protein levels are higher in CRYAB^R120G transgenic mice with hypertrophy.**

(A) Heart weight-to-body weight ratio (HW/BW) in 1-, 4- and 7-month-old CRYAB^R120G transgenic (R120G) and non-transgenic (NTG) mice. (B) Echocardiography. Left ventricular ejection fraction (LVEF) in 7-month-old R120G and NTG mice. (C) Representative Western blots of phosphorylated (P-)STAT3, STAT3 and indicated control in 1-, 4- and 7-month-old R120G and NTG mice. (D) P-STAT3/STAT3 quantification of Western blots from 1-, 4- and 7-month-old R120G and NTG mice. (E) Stat3 mRNA level determined by RT-qPCR from 7-month-old R120G and NTG mice. (F) Western blot of harvested hearts after fractionation into soluble and insoluble with respective quantification. WB quantification with Image Studio™ or Image Lab™ or Bio-1D-Vilber software. Data were obtained with a minimum of 3 replicates per condition. The number of replicates is indicated by the dots. Data are depicted as mean ± SEM, and p-values were obtained with the unpaired Student's t-test. Source data are available online for this figure.

LVIDd, and LVM/BW were not markedly affected by *Jak1* KO (Fig. 10D). Cardiac hypertrophy represented by HW and HW/BW was higher in DRM than in NTG mice with *Jak1* wt and unaffected by *Jak1* KO (Fig. 10E). Body weights did not differ between the groups (Fig. 10E). As expected, P-STAT3 levels were lower in R120G mice with *Jak1* KO than with *Jak1* wt (Fig. 10F). CRYAB^R120G aggregate load was lower in *Jak1* KO than in *Jak1* wt mouse ventricular tissues, determined by immunofluorescence (Fig. 10G) and Western blot (Fig. 10H). Overall desmin structure was improved by ruxolitinib treatment (Fig. 10I, indicated by green arrows).

## Discussion

In the present study, we tested the impact of JAK1 knockdown and inhibition in CRYAB^R120G DRM models. Our main findings were: (i) ruxolitinib treatment prevented the formation of and cleared CRYAB^R120G aggregates in NRVMs and hiPSC-CMs; (ii) knockdown of *Jak1* and *Stat3* prevented the formation of CRYAB^R120G aggregates; (iii) ruxolitinib treatment resulted in higher UPS function and higher RNA levels of muscle-relevant E3 ubiquitin ligases including ASB2; (iv) CRYAB bound to ASB2, accumulated and was less ubiquitinated if inactive ASB2 was expressed; (v) blocking UPS function or knocking down *Asb2* blunted the effect of ruxolitinib on CRYAB^R120G aggregates and v) JAK1 loss-of-function prevented cardiac dysfunction in DRM mice.

Ruxolitinib, first FDA-approved in 2011, is used to treat myelofibrosis, polycythemia vera, and graft versus host reaction in allogeneic stem cell transplantation. Furthermore, repurposing of ruxolitinib is tested for the treatment of COVID and autoimmune diseases (Hammersen et al, 2023; Levy et al, 2023). Ruxolitinib competitively inhibits the ATP-binding site of JAK1 and JAK2 (IC$_{50}$ = 3.3 nM and 2.8 nM in cell-free assay, respectively (Ostojic et al, 2011). Reported adverse events with ruxolitinib predominantly include infections, and inhibition of hematopoiesis, thus, a regular check of blood count and kidney function is necessary (Lussana et al, 2018; Polverelli et al, 2017). Although ruxolitinib was approved in 2011, it is still prescribed to a limited patient population, and off-label uses are not well-documented. Ruxolitinib is approved for long-term use in conditions like myelofibrosis and polycythemia vera, where it can be used for several years (Aruga et al, 2025; Kiladjian et al, 2020; Verstovsek et al, 2017). However, due to its immunosuppressive effects and hematological toxicity, close monitoring for infections and blood count abnormalities is crucial. In our study, we looked closer into the cardiotoxicity of ruxolitinib and treated hiPSC-derived EHTs with concentrations ranging from 0.3–30 µM over several weeks. Reported patient plasma concentrations of ruxolitinib range from around 0.2 to 1.7 µM, depending on the dosage (Appeldoorn et al, 2023). Minor toxicity effects were observed in EHTs with concentrations around 30-fold higher than reported in patients' plasma. In the future, other inhibitors, such as those targeting STAT3, may also be of interest. Ultimately, the use of ruxolitinib for DRM caused by *CRYAB* variants may constitute off-label treatment, depending on the severity of the DRM progression and the patient's ability to tolerate the drug. In the past years, there has been a race to identify JAK inhibitors which are more specific to one member of the family, though with limited success. In this study, we tested 3 inhibitors, solcitinib (not approved), upadacitinib (FDA-approved), and filgotinib (FDA-approved), which were primarily tested for inhibiting mutant JAK1 in diseases such as rheumatoid arthritis or inflammatory bowel disease. We found that all of them efficiently reduced CRYAB^R120G aggregate load in NRVMs and hiPSC-CMs. Since none of the available inhibitors is entirely specific to JAK1, we performed siRNA knockdown experiments. We found that *Jak1* knockdown prevented CRYAB^R120G aggregate accumulation and that ruxolitinib treatment of siJak1-treated cells did not further increase UPS function, highlighting the dependence on JAK1 for this effect. In addition, *Jak1* and *Stat3* mRNA levels were markedly higher than the other JAK and STAT members in NRVMs and hiPSC-CMs, suggesting their important role in cardiomyocytes. Especially, *Jak3*, *Stat4*, *Stat5a*, and *Stat5b* mRNA levels were very low or not detected in NRVMs and hiPSC-CMs and may, therefore, be less important than the other JAK-STAT family members in cardiomyocytes. In line with this, whole-body knockouts of *Jak1*, *Jak2*, and *Stat3* are lethal in mice, whereas knockouts of the other JAK-STAT pathway members are viable with defects in immune response and/or hematopoiesis (Rodig et al, 1998).

Variants in JAKs can cause severe diseases such as rheumatoid arthritis and inflammatory bowel diseases for *JAK1* variants, myeloproliferative neoplasms for *JAK2* variants, and severe immune deficiency for *JAK3* and *TYK2* variants (Luo et al, 2021). Similarly, several *STAT* variants increase the incidence of malignancies, infections, and autoimmune diseases (O'Shea et al, 2015). The role of JAK1 in cardiac health and disease remains relatively obscure. For STAT3, several studies pointed out its importance in the heart. For example, cardioprotective roles have been described, such as in cardiomyocyte-specific *Stat3* knock-out in mice, which were more susceptible to ischemia/reperfusion injury, had reduced cardiac function, and increased mortality after myocardial infarction and with age (Hilfiker-Kleiner et al, 2004; Jacoby et al, 2003). On the other hand, STAT3 has been described to stimulate cardiac hypertrophy in experimental models (Kunisada et al, 2000; Kunisada et al, 1998). However, the role of STAT3 in cardiomyocyte hypertrophy was questioned when interleukin-6

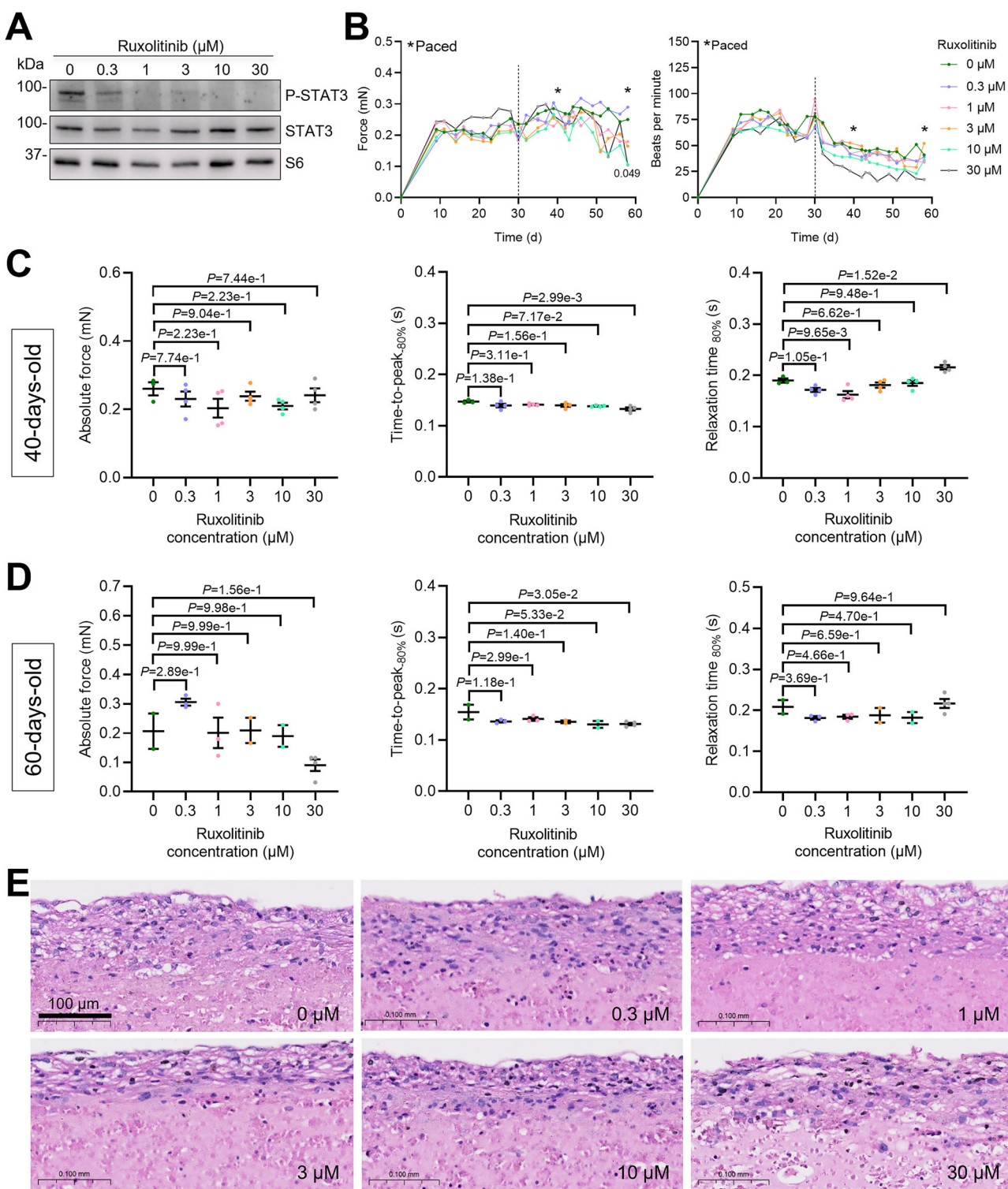

**Figure 8.  Evaluation of potential cardiotoxic effects of ruxolitinib in human engineered heart tissues (EHTs).**

(A) Western blot of EHTs treated with different concentrations of ruxolitinib stained for phosphorylated STAT3 (P-STAT3), STAT3, and S6 as a loading control. (B) Time-course measurements of contractile force and beating frequency in EHTs. Asterisks indicate days on which EHTs were electrically paced. (C) Measurements of average peak, force amplitude, time to 80% of peak (time-to-peak$_{-80\%}$), and time to 80% relaxation (relaxation time$_{80\%}$) of EHTs electrically paced on day 40, and 60. (D) Histological evaluation of 60-day-old EHTs. Hematoxylin-eosin (H&E) staining following 4 weeks of chronic ruxolitinib treatment. Data are depicted as mean ± SEM, and p-values were obtained with the one-way ANOVA and Dunnett's multiple comparisons post-hoc analysis (C). Source data are available online for this figure.

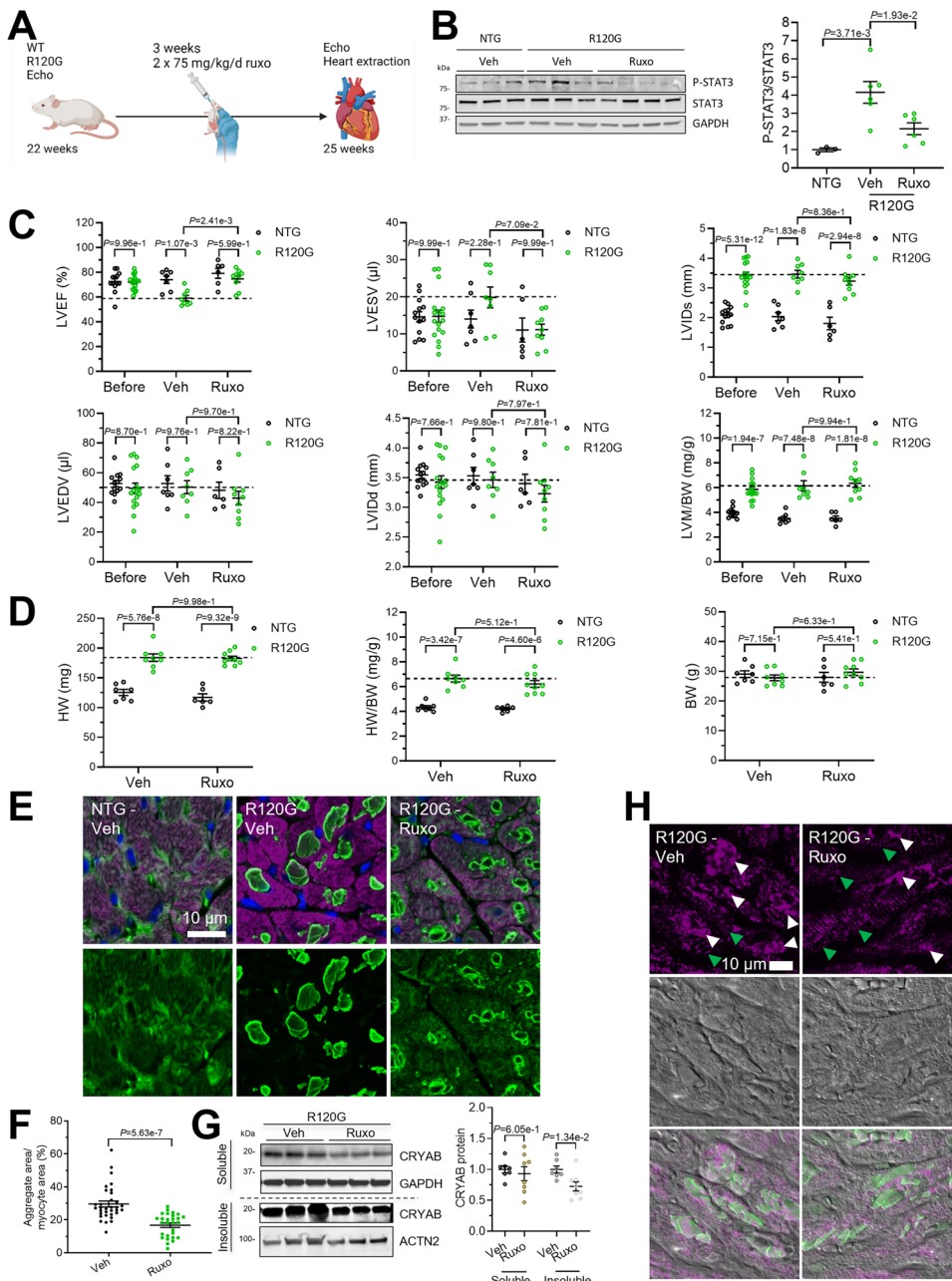

**Figure 9.   Ruxolitinib treatment of CRYAB^R120G transgenic mice prevents the development of cardiac dysfunction.**

CRYAB^R120G transgenic (R120G) and non-transgenic (NTG) mice treated for 3 weeks with 75 mg/kg ruxolitinib (ruxo) or vehicle (veh) twice-daily oral gavage. Transthoracic echocardiography was performed at the start (before, 21-week-old) and the end of the treatment (veh or ruxo, 24-week-old). (A) Scheme of experimental outline. (B) WB and quantification of phosphorylated (P-) STAT3, STAT3 and indicated controls at the end of treatment. WB quantification was performed with Image Lab software. For NTG mice $n = 3$, for R120G Veh and Ruxo mice $n = 6$. (C) Echocardiography. Left ventricular ejection fraction (LVEF), left ventricular end-systolic volume (LVESV), left ventricular internal diameter systole (LVIDs), left ventricular end-diastolic volume (LVEDV), left ventricular internal diameter diastole (LVIDd), and left ventricular mass-to-body weight ratio (LVM/BW) before and after (veh/ruxo) treatment. (D) Heart weight (HW), heart weight-to-body weight ratio (HW/BW) and body weight (BW) at the end of the treatment. (E) Representative images (F), and quantification of R120G and NTG mouse heart sections after vehicle or ruxolitinib treatment. CRYAB is depicted in green, cardiomyocytes in purple (anti-cardiac troponin I) and nuclei in blue (DAPI). Scale bar = 10 μm. Quantification of aggregates in cardiomyocytes with NIS Elements software. At least 5 images of 3 mice per group were analyzed. (G) Western blot after fractionation into soluble and insoluble, and quantification of CRYAB. (H) Representative images of R120G and NTG mouse heart sections after vehicle or ruxolitinib treatment. Desmin is depicted in purple, and CRYAB in green. Scale bar = 10 μm. Green arrows indicate aligned desmin structure. White arrows depict desmin aggregates. Data are depicted as mean ± SEM, and p-values were obtained with the one-way (B) or two-way ANOVA and Sidak's or Tukey's (comparison of R120G groups) multiple comparisons post-hoc analysis (C, D) or unpaired Student's t-test (F, G). Source data are available online for this figure.

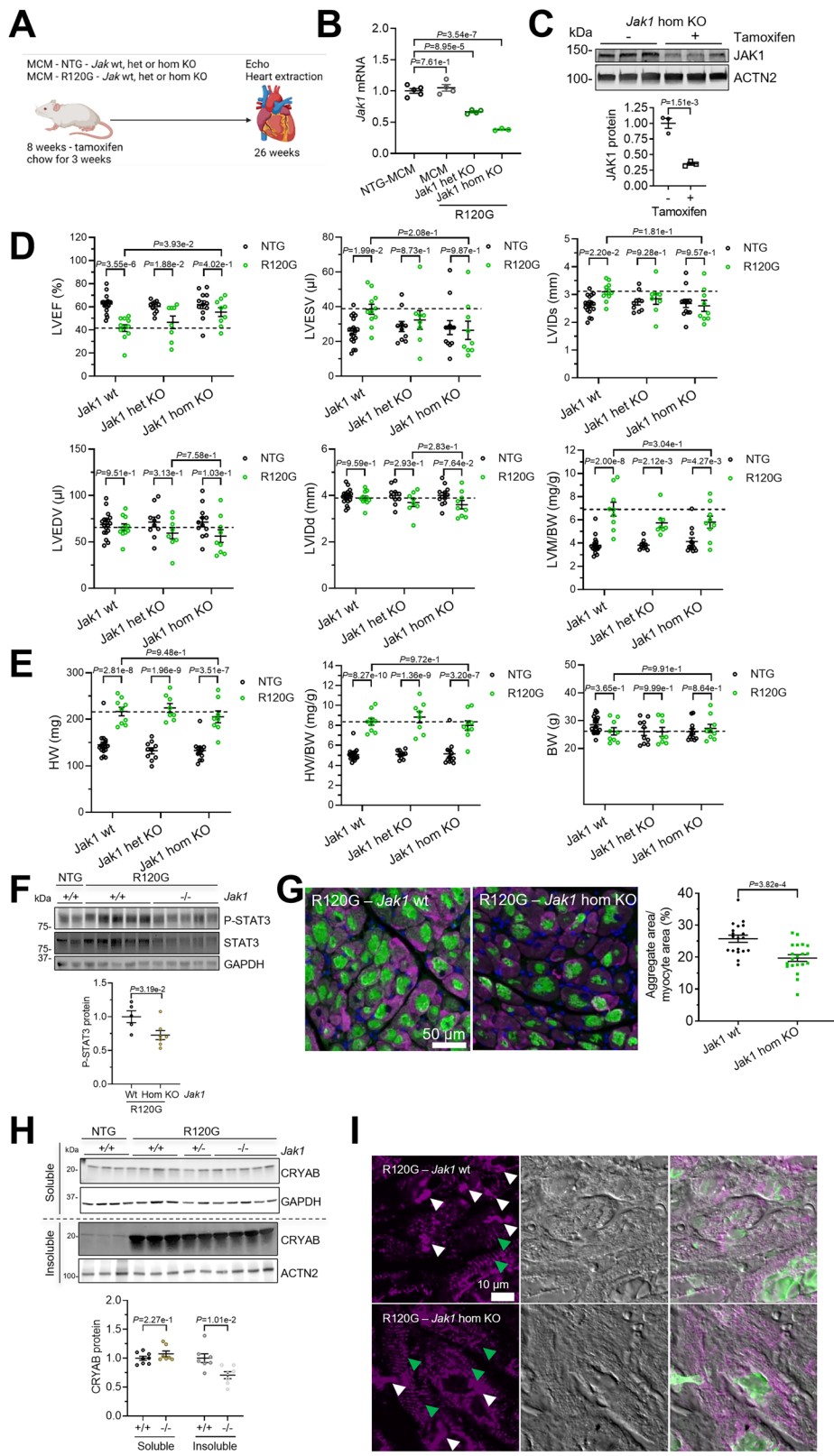

Figure 10. *Jak1* knockout prevents development of cardiac dysfunction in CRYAB[R120G] transgenic mice.

Heterozygous (het) or homozygous (hom) *Jak1* knockout (KO) was induced in MCM-transgenic (TG) mice crossed with CRYAB p.Arg120Gly TG (R120G) or non-transgenic (NTG) mice with tamoxifen chow. Transthoracic echocardiography was performed 26 weeks and hearts were extracted. (A) Scheme of experimental outline. (B) *Jak1* mRNA levels determined by RT-qPCR. (C) JAK1 protein levels determined by Western blot and normalized to ACTN2. Western blot quantification was performed with Image Lab software (D), Echocardiography. Left ventricular ejection fraction (LVEF), left ventricular end-systolic volume (LVESV), left ventricular internal diameter systole (LVIDs), left ventricular end-diastolic volume (LVEDV), left ventricular internal diameter diastole (LVIDd), and left ventricular mass-to-body weight ratio (LVM/BW). (E) Heart weight (HW), heart weight-to-body weight ratio (HW/BW) and body weight (BW). (F) Western blot and quantification of indicated proteins. (G) Representative images and quantification of *Jak1* KO and *Jak1* WT mouse heart sections. Aggregates are depicted in green, cardiomyocytes in purple (anti-cardiac troponin I) and nuclei in blue (DAPI). Scale bar = 50 μm. Quantification of aggregates in cardiomyocytes with NIS Elements software. At least 5 images of 3 mice per group were analyzed. (H) Western blot of indicated proteins after fractionation into soluble and insoluble, and quantification of CRYAB. (I) Representative images and quantification of *Jak1* KO and *Jak1* WT mouse heart sections. Desmin is depicted in purple, and CRYAB in green. Scale bar = 10 μm. Green arrows indicate aligned desmin structure. White arrows depict desmin aggregates. Data are depicted as mean ± SEM, and *p*-values were obtained with the one-way (B) or two-way ANOVA (D, E) with Sidak's or Tukey's (comparison of R120G groups) multiple comparisons post-hoc analysis or unpaired Student's t-test (C, F–H). Source data are available online for this figure.

deletion, reducing STAT3 signaling, did not affect the development of hypertrophy (Gonzalez et al, 2015). In human hearts, higher (Ng et al, 2003) or lower (Podewski et al, 2003) P-STAT3 levels were observed in dilated cardiomyopathy. Our study found that the steady-state level of P-STAT3 and JAK1 was higher in DRM mice. Another recent study found higher steady-state levels of P-STAT1 in DRM mice (Cai et al, 2024), supporting our findings.

Apart from the canonical JAK-STAT signaling, a few non-canonical functions have been proposed for JAK1. For example, direct epigenetic regulation of gene expression by histone H3 phosphorylation through JAK has been described (Rui et al, 2016). Furthermore, one study showed that JAK1 regulates many genes in response to ER stress and co-immunoprecipitates with ATF4 (Sims and Meares, 2019). Yet another study found that ER stress stimulated JAK1-dependent STAT3 activation (Meares et al, 2014). Here, we provide evidence for a novel function of JAK1-STAT3 in regulating the UPS, which may pave the way for yet unconsidered applications of JAK1 inhibitors. The JAK-STAT pathway is an important regulator of gene expression, and we found that *Jak1* knockdown led to higher expression of the E3 ligase ASB2. ASB2 is a member of the SOCS (suppressor of cytokine signaling)–box family, many of which are known direct transcriptional targets of STATs (Anasa et al, 2018). JAK–STAT signaling may also influence *Asb2* expression by modulating other transcription factors that, in turn, regulate the *Asb2* locus. In addition, ASB2 has been shown to promote degradation of JAK2 and JAK3 (Nie et al, 2011; Wu and Sun, 2011), suggesting that they could participate in a feedback loop in which JAK–STAT signaling regulates *Asb2* expression. ASB2 has been shown previously to target desmin and filamins (Heuze et al, 2008; Razinia et al, 2011; Thottakara et al, 2015) for proteasomal degradation, proteins that also accumulate in aggregates with CRYAB (Feldkirchner et al, 2012). We found that CRYAB also accumulates, co-immunoprecipitates, and is less ubiquitinated with the overexpression of mutant ASB2β, suggesting that it is also a substrate of ASB2β. CRYAB was also significantly enriched in a previous ASB2β interactome and ubiquitinome study (Goodman et al, 2021), strengthening our hypothesis. Therefore, we hypothesize that in our model, JAK1 loss-of-function results in higher levels of ASB2β, which then increasingly targets desmin and CRYAB for proteasomal degradation, thus resulting in fewer aggregates. The UPS is responsible for the specific physiological and pathological degradation of many proteins, including desmin and CRYAB (Zhang et al, 2010). For a long time, it has been assumed that the UPS can only

remove single proteins, but not larger protein aggregates. However, it has been shown that the UPS removes protein aggregates in the nucleus, where there is no access to the autophagic machinery (Iwata et al, 2009), and in the cytosol of autophagy-defective *Atg5* knockout cells (Hjerpe et al, 2016). In our experiments, we found a mild reduction of the autophagy-lysosomal pathway and chaperone-mediated autophagy after ruxolitinib treatment, suggesting that the UPS is responsible for clearing the CRYAB[R120G] aggregates. Several studies have shown that the upregulation of UPS or ALP function can prevent the formation of CRYAB[R120G] aggregates (Pan et al, 2017; Xu et al, 2020; Yang et al, 2023) but did not show the clearance of pre-existing aggregates. This is important because the aggregates are already present when patients develop symptoms and seek health care. Importantly, our experiments suggest that ruxolitinib can clear small and medium-sized pre-existing CRYAB[R120G] aggregates. In the cells where dissolution is observed after ruxolitinib treatment, the aggregates do not reappear for the length of our conducted experiments (5 days). We observed that cells are differently affected by ruxolitinib treatment, and hypothesize that there is a certain aggregate threshold. Beyond this threshold, aggregates cannot be cleared anymore. In our videos, it is evident that all cardiomyocytes with smaller aggregates can clear aggregates after ruxolitinib treatment, whereas in cells with larger aggregates, some cells can clear them, while others cannot. This leads us to the assumption that cells with larger aggregates have either not reached the threshold yet and the aggregates can still be cleared, or are beyond the threshold, where they cannot be cleared anymore. It is also possible that the aggregation of other proteins, such as desmin, plays a role in this context. A potential treatment might have to start relatively early in the disease. We intentionally started our experiments in mice at a late time point, at 5 months of age for ruxolitinib treatment and at 2 months of age for induced *Jak1* knockout, to explore its translational impact. They revealed that targeting JAK1 after formation of aggregates can still prevent a drop in cardiac function, suggesting that a relatively late start of treatment would still be of significant benefit. Interestingly, the ruxolitinib treatment at 5 months and the *Jak1* cardiomyocyte-specific knockout at 2 months had similar effects on cardiac function and aggregate load, leading to the assumption that a late intervention may be similarly effective as an early intervention. However, a systematic evaluation, importantly in several DRM models, would be necessary to answer this question completely.

Nevertheless, ruxolitinib treatment or *Jak1* cardiomyocyte-specific KO in DRM mice underlined its therapeutic potential in vivo since both led to lower CRYAB$^{R120G}$ aggregates and prevented a drop in cardiac function. Cardiac hypertrophy was unaffected, which aligns with the finding that hypertrophy does not differ between 4- and 7-month-old DRM mice, although the aggregate load increases. There may be a certain threshold of aggregate load to establish hypertrophy. In our results, ruxolitinib treatment or cardiomyocyte-specific *Jak1* KO led to similar effects on cardiac function, suggesting that the observed effects are caused by *Jak1* KO in cardiomyocytes. An intriguing question to answer in future studies is whether other mechanisms, such as inhibition of cardiac fibrosis (Meng et al, 2024) or a potential inflammatory response, add to the protective effect of ruxolitinib in DRM mice.

Important limitations that remain in this study include that the precise mechanism of *Asb2* expression regulation is still unknown, and that it is unclear whether ruxolitinib is also effective on other genetic variants causing protein aggregates, currently limiting its effect to CRYAB aggregates.

In conclusion, it will be of great interest to further test if repurposing of JAK1 inhibitors is a potential therapy for DRM caused by CRYAB$^{R120G}$. Also, testing the effect of ruxolitinib or other more specific JAK1 inhibitors in different DRM models and other protein aggregate-driven diseases would be important.

# Methods

## Reagents and tools table

| Reagent/Resource | Reference or Source | Identifier or Catalog Number |
|---|---|---|
| **Experimental methods** | | |
| Sprague Dawley or Wistar neonatal rats | Envigo | |
| UKEi001 | Pietsch et al, 2023 | ERC001 |
| mTagRFPT-TUBA1B | Coriell Institute | AICS-0031-035 |
| Cardiomyocyte-specific (Myh6 promoter) transgenic CRYABR120G | Wang et al, 2001 | |
| Conditional Jak1 knockout (KO) C57BL/6 mice | Taconic Biosciences | |
| Cardiomyocyte-specific (Myh6 promoter) MerCreMer (MCM) mice | Sohal et al, 2001 | |
| RNA polyA-stranded library, paired-end 75 bp sequencing conditions | | |
| **Recombinant DNA** | | |
| AdV5-CRYABR120G-GFP | Singh et al, 2021 | |
| AdV5-GFP | Singh et al, 2021 | |
| AdV5-GFPu | Singh et al, 2021 | |
| AdV5-Flag-Asb2 WT | Thottakara et al, 2015 | |
| AdV5-Flag-Asb2 Mut | Thottakara et al, 2015 | |
| **Antibodies** | | |
| TNNI3 (troponin I, cardiac 3) | Millipore | MAB1691 |

| Reagent/Resource | Reference or Source | Identifier or Catalog Number |
|---|---|---|
| ACTN2 | Sigma-Aldrich | A7811 |
| DAPI (nuclei) | ThermoFisher Scientific | D1306 |
| Phalloidin (Phalloidin-Alexa Fluor 633) | ThermoFisher Scientific | A22284 |
| ProLongTM Gold antifade | ThermoFisher Scientific | P36930 |
| CRYAB | Stressgen | SPA-222 |
| SiR-actin | Spirochrome | SC001 |
| SPY-tubulin | Spirochrome | SC503 |
| **Oligonucleotides and other sequence-based reagents** | | |
| scramble siRNA | ThermoFisher Scientific | 4390846 |
| siJak1 siRNA | ThermoFisher Scientific | s7646-48 |
| siJak2 siRNA | ThermoFisher Scientific | s76450 |
| siStat3 siRNA | ThermoFisher Scientific | s129048 |
| siPsmd1 siRNA 20 nM | ThermoFisher Scientific | s136286 |
| siAsb2 siRNA | ThermoFisher Scientific | s28515 |
| siRapsn siRNA | ThermoFisher Scientific | s168618 |
| siRnf207 siRNA | ThermoFisher Scientific | 193639 |
| siTrim50 siRNA | ThermoFisher Scientific | s143615 |
| CMV promoter | | |
| **Chemicals, Enzymes, and other reagents** | | |
| Calicum and Bicarbonate-free Hanks with HEPES (CBFHH) buffer | Roth | 9105.4 |
| Trypsin | Difco | 0152-15-9 |
| DNase/FCS (active) solution | Sigma | D8764 |
| DMEM | Gibco | 52100-021 |
| FCS | ThermoFisher Scientific | A5670901 |
| Penicillin-Streptomycin (P/S) | Gibco | 15140-122 |
| DMEM F-12 | Gibco; Life Technologies | 21331046 |
| L-glutamine | Gibco; Life Technologies | 25030024 |
| Transferrin | Sigma-Aldrich; Merck | T8158-1G |
| Selenium | Sigma-Aldrich; Merck | S5261 |
| Human Serum Albumin | Biological Industries | 05-720-1B |
| Lipid Mixture | Sigma-Aldrich, Merck | L5146 |

| Reagent/Resource | Reference or Source | Identifier or Catalog Number |
| --- | --- | --- |
| Insulin | Sigma-Aldrich, Merck | I9278 |
| Dorsomorphin | Selleckchem | S7306 |
| rh Activin A ACF | STEMCELL Technologies | 78132.2 |
| TGF-β1 | Peprotech | 100-21 C |
| Human Recombinant b-FGF | Peprotech | 100-18B |
| GeltrexTM 375 | Gibco | A14133-02 |
| Activin A/bone morphogenic factor (BMP4) | R&D Systems, Life Technologies | 314-BP |
| Horse serum | Gibco | 26050088 |
| Aprotinin | Genaxxon | M6361.1010 |
| Gelatine | Sigma-Aldrich | G1393 |
| OptiMEM | ThermoFisher Scientific | 51985-026 |
| Lipofectamine3000 | ThermoFisher Scientific | L3000008 |
| FBS | Biochrom | S0615 |
| cytosine β-D-arabinofuranoside | Sigma-Aldrich | C1768 |
| Adenovirus serotype 5 (AdV5) | In-house Vector Facility | |
| DMSO | Sigma-Aldrich | D4540 |
| Ruxolitinib | Selleckchem | S1378 |
| Filgotinib | Selleckchem | S7605 |
| Solcitinib | Selleckchem | S5917 |
| Upadacitinib | Selleckchem | S8162 |
| Bafilomycin A1 | Sigma-Aldrich | B1793 |
| Torin 2 | LC laboratories | T8448 |
| ROTI® Histofix | Carl Roth | P087.1 |
| RNAzol RT Reagent | Molecular Research Center Inc. | RN 190 |
| iScript cDNA synthesis kit | Bio-Rad | 17,08,840 |
| TaqMan gene expression assay - Psmd1 | ThermoFisher Scientific | Rn01400483_m1 |
| TaqMan gene expression assay - Jak1 | ThermoFisher Scientific | Rn01400483_m1 |
| TaqMan gene expression assay - Jak2 | ThermoFisher Scientific | Mm00600614_m1 |
| TaqMan gene expression assay - Stat1 | ThermoFisher Scientific | Mm01208489_m1 |
| TaqMan gene expression assay - Stat3 | ThermoFisher Scientific | Mm01219775_m1 |
| TaqMan gene expression assay - Tyk2 | ThermoFisher Scientific | Mm00444469_m1 |
| TaqMan gene expression assay - 18 s | ThermoFisher Scientific | Hs03003631_g1 |
| SsoAdvancedTM Universal Probes Supermix | BioRad | 17,25,281 |
| CellLytic M | Sigma-Aldrich | C2978 |

| Reagent/Resource | Reference or Source | Identifier or Catalog Number |
| --- | --- | --- |
| SDS | Carl Roth | 205-788-1 |
| Tris base | Sigma-Aldrich | 3,36,028 |
| EDTA | Carl Roth | 205-358-3 |
| NaF | Sigma-Aldrich | MKBR0679V |
| Glycerol | Sigma; Merck | G5516 |
| DTT | Sigma; Merck | D9779 |
| Protease inhibitor cocktail (cOmpleteTM mini, EDTA-free) | Roche | 4,69,31,24,001 |
| Protease inhibitor cocktail (PhosSTOPTM) | Roche | PHOSS-RO |
| 6x Laemmli buffer | ThermoFisher Scientific | J60660.AC |
| Methanol | House-made | |
| PFA | ThermoFisher Scientific | 047340.9 M |
| PBS | Gibco | 10010-049 |
| Triton X100 | Sigma-Aldrich | STBJ4510 |
| BSA | Serva | 11930.03 |
| Tween 20 | Sigma-Aldrich | SLCL0578 |
| Sodium Azide | Sigma-Aldrich | STBJ1309 |
| **Software** | | |
| Visual Sonics Vevo 2100 Imaging System (40 Hz transducer) | | |
| NanoString | | |
| Image Studio | LICOR | |
| Image Lab | Bio-Rad | |
| Bio-1D (Vilber) | | |
| Nikon Eclipse Ti inverted microscope equipped with an Andor Zyla-4.2 sCMOS camera, using a Plan Apo ʎ 20x/0.75 objective lens | | |
| Zeiss Axio Observer.Z1/7 microscope LSM800 with a Plan-Apochromat 20x/0.80 objective lens | | |
| Nikon Eclipse Ti2-E inverted microscope equipped with an Okolab incubation chamber | | |
| Nikon BioStation IM-Q time-lapse imaging system with a 40x/0.80 objective lens | | |
| NIS Elements software | | |
| ImageJ | | |
| GraphPad Prism 8 | | |
| **Other** | | |
| 6-well culture plates | Nalge Nunc International | 154461 |
| 12-well culture plates | Nalge Nunc International | 154461 |
| Lab-Teks | Nalge Nunc International | 154461 |

| Reagent/Resource | Reference or Source | Identifier or Catalog Number |
|---|---|---|
| 6-well plates with nanopattern coverslips | Tebu-Bio | ANFS-CS25 904 |
| 96-well plates | ThermoFisher Scientific | 165305 |
| 4–15% or 4–20% Mini-Protean TGX™ Precast Protein Gels +XFDA67:C68 | Bio-Rad | 45,61,084 |
| 8 or 12% self-casted acrylamide/ bisacrylamide (29:1) gels | House-made | |
| Nitrocellulose Membranes | BioRad | BR20240117 |
| Silicone racks | EHT Technologies Hamburg | |
| Teflon spacers | EHT Technologies Hamburg | |
| EHT analysis device - White Box | EHT Technologies Hamburg | |
| EHT analysis software - CTMV | EHT Technologies Hamburg | |

## Isolation of NRVMs

Isolation of hearts from neonatal rats was performed by decapitation without anesthetic agents and conformed to the guidelines from directive 2010/63/EU of the European Parliament on the protection of animals used for scientific purposes, the corresponding German authorities, and the NIH Guide for Care and Use of Laboratory Animals (for isolations done in the USA). Hearts were harvested from 1–3-day-old pups (Sprague Dawley or Wistar), washed in Calcium and Bicarbonate-free Hanks with HEPES (CBFHH) buffer, cut into small pieces, and dissociated 3–5 min step-wise with 0.25% trypsin (Difco, 0152-15-9) at RT. Dissociated cells were collected in DNase (Sigma, D8764)/FCS (active) solution, spun down at $60 \times g$ and 4 °C for 15 min, resuspended in medium (DMEM, 10% FCS, 1% P/S), and dropped through a cell strainer. Cells were then pre-plated for 60–90 min to reduce the number of non-cardiomyocytes in the supernatant.

## HiPSC culture, cardiomyocyte differentiation, and engineered heart tissues (EHTs)

UKEi001-A (Pietsch et al, 2023) (ERC001) or mTagRFP-TUBA1B (AICS-0031-035, Coriell Institute; for JAK-STAT pathway expression values) hiPSCs were expanded and differentiated using either embryoid body and growth factor-based three-stage protocol (Breckwoldt et al, 2017) or a small-molecule-based protocol according to Pietsch et al (2024). In brief, the expansion of the hiPSCs was performed in FTDA medium (house-made, DMEM F-12, L-glutamine 2 mM, transferrin 5 µg/ml, selenium 5 ng/ml, human serum albumin 0.1%, lipid mixture 1x, insulin 5 µg/ml, dorsomorphin 50 nM, activin A 2.5 ng/ml, TGFβ1 0.5 ng/ml, bFGF 30 ng/ml) on Geltrex™-coated (Gibco A14133–02) cell culture vessels. Cardiomyocyte differentiation was achieved by treating

high-density, undifferentiated monolayer iPSC cultures with an activin A/bone morphogenetic protein 4 (BMP4)-based protocol. After dissociation, hiPSC-CMs were seeded at a density of $5 \times 10^6$ cells/well in complete medium (house-made, DMEM, penicillin-streptomycin 1%, horse serum (inactive) 10%, insulin 10 µg/ml, aprotinin 33 µg/ml; Sigma, A1153) on Geltrex™ (Gibco A14133–02)-coated 6-well culture plates (Nalge Nunc International, 154461) and kept at 37 °C and 7% $CO_2$ with medium change on Mondays, Wednesdays and Fridays.

For engineered heart tissues (EHTs), cardiomyocyte cell suspensions were cast into a 3D format in a 24-well plate. Each EHT composition is as follows: $1 \times 10^6$ hiPSC-CMs/EHT in a fibrin matrix (total volume 100 µL) consisting of 5 mg/ml bovine fibrinogen (200 mg/ml in NaCl 0.9%, 0.5 µg/mg aprotinin (Sigma, A1153), 2× DMEM, 10 µmol/l Y-27632, and 3 U/ml thrombin (Biopur, BP11101104)), as described previously (Breckwoldt et al, 2017). Silicone racks (C001), Teflon spacers (C002), and EHT analysis equipment (A001) were purchased from EHT Technologies GmbH (Hamburg, Germany). EHTs were cultured at 37 °C in 7% $CO_2$ and 40% $O_2$, and medium exchange was performed every Monday, Wednesday, and Friday with complete medium.

## siRNA transfection, compound treatment, and virus transduction of cardiomyocytes

NRVMs were seeded into 0.1% gelatine-coated (Sigma-Aldrich, G1393) 12- or 6-well plates, Lab-Teks (Nalge Nunc International, 154,461) or 6-well plates with nanopattern coverslips (Tebu-Bio ANFS-CS25 904). Per 12-, 6-well plate, or LabTek well, $1.5 \times 10^5$, $3 \times 10^5$, or $2 \times 10^5$ cells were seeded, respectively. HiPSC-CMs were seeded on Geltrex™-coated 6-well plates with nanopattern coverslips or 96-well plates (ThermoFisher Scientific, 165305). Per 6-well- or 96-well plate well, $3 \times 10^5$ or $1 \times 10^4$ cells were seeded, respectively. For siRNA transfection of NRVMs, the medium was changed to 500 µL (12-well and LabTeks) or 1 mL (6-well) OptiMEM (Thermo Fisher Scientific, 51985-026). SiRNAs and Lipofectamine3000 (Thermo Fisher Scientific, L3000008) transfection reagent (final amount of 4 µL/mL) were diluted in 50 µL (12-well and LabTeks) or 100 µL (6-well) OptiMEM each. After 5 min of incubation, the solutions were mixed and incubated for 20 min before adding dropwise to the wells. Four hours later, DMEM with 20% FBS was added vol/vol, or cells were transduced with adenovirus. Two days later, the medium was changed to culture medium with 10 µM cytosine β-D-arabinofuranoside (Sigma-Aldrich, C1768). Following siRNAs were used as indicated: scramble siRNA (corresponding concentration, Thermo Fisher Scientific, 4390846), siJak1 (Thermo Fisher Scientific, s7646-48), siJak2 (Thermo Fisher Scientific, s7650), siStat3 (Thermo Fisher Scientific, s129048), siPsmd1 (20 nM; Thermo Fisher Scientific, s136286), siAsb2 (Thermo Fisher Scientific, s28515), siRapsn (Thermo Fisher Scientific, s168618), siRnf207 (Thermo Fisher Scientific, s193639), or siTrim50 (Thermo Fisher Scientific, s143615).

For overexpression of CRYAB$^{R120G}$-GFP, GFP, GFPu, Flag-ASB2β WT or Flag-ASB2β L595A, NRVMs or hiPSC-CMs were transduced with an adenovirus serotype 5 (AdV5) under the control of a CMV promoter. The appropriate virus amount (MOI 0.5–10) was determined in preliminary experiments and varied between virus productions. The hiPSC-CMs were transduced with

about double the MOI compared to NRVMs. For transduction, the virus was diluted in 500 μL (12-well and LabTeks) or 1 mL (6-well) OptiMEM, and the medium was replaced by the solution for 2 h. If cells were transfected before transduction, the virus was diluted in 250 μL (12-well and LabTeks) or 500 μL (6-well) OptiMEM and added to the transfection solution after 4 h. After 2 h of virus transduction, DMEM with 20% FBS (NRVMs) or double-concentrated hiPSC-CM culture medium was added vol/vol.

All compounds were dissolved in DMSO. The final concentration of DMSO was 0.1%. The JAK inhibitors ruxolitinib (3 μM), filgotinib (1 μM), solcitinib (1 μM), and upadacitinib (1 μM) were added either from day 1 with the double-concentrated medium) or as indicated. For live cell imaging, 10 μM ruxolitinib or 0.1% DMSO was added once right before the experiment. The culture medium with inhibitors was refreshed every other day. For autophagic flux measurement, cells were treated with 50 nM bafilomycin $A_1$ (Sigma-Aldrich, B1793) and/or 50 nM torin 2 (LC Laboratories, T8448) for 3 h.

## Analysis of contractile force in EHTs and inhibitor treatment with ruxolitinib

Contractile force in EHTs was assessed. Briefly, measurements were performed every Monday, Wednesday, and Friday inside a transparent chamber using an automated video-optical system that recorded the deflection of silicon posts at a homeostatic of 37 °C and 5% $CO_2$. Determination of contractile force was analyzed based on the known mechanical properties of the silicon using a customized software (CTMV, Pforzheim, Germany. After 30 days in culture, EHTs were subjected to Ruxolitinib for 4 weeks at different concentrations (0, 0.3, 1, 3, 10, 30 μM) with the compound being renewed with every medium exchange. EHTs were electrically paced for experimental analysis (1 V, 2 Hz, impulse duration 4 ms). The contraction peaks were analyzed for contraction time (time-to-peak$_{80\%}$; TTP$_{80\%}$), and relaxation time (RT$_{80\%}$) at 80% of peak height, and force amplitude. For histology, EHTs are fixed overnight in Histofix (Carl Roth) at 4 °C while still being attached to the silicone rack to ensure stretched sarcomeres. Fixed tissues are carefully removed from the posts of the silicone racks with forceps and stored in phosphate-buffered saline in closed vessels of appropriate size at 4 °C until further processing. Embedding, sectioning and immunohistochemistry were performed by Kristin Hartmann (UKE core facility, Mouse Pathology).

## Mouse experiments

The investigation conformed to the guide for the care and use of laboratory animals published by the National Institutes of Health (Publication No. 85-23, revised 2011, published by the National Research Council). Both sexes were used for the studies. Cardiomyocyte-specific (Myh6 promoter) transgenic CRYAB[R120G] (Wang et al, 2001) were maintained on FVBN background. Conditional Jak1 knockout (KO) C57BL/6 mice were purchased from Taconic Biosciences (C57BL/6-Jak1[tm1.1 mrl]) and crossed with CRYAB[R120G] and cardiomyocyte-specific (Myh6 promoter) Mer-CreMer (MCM) mice (Sohal et al, 2001) on FVBN background for at least five generations. Mice were sacrificed by cervical dislocation under isoflurane or $CO_2$ anesthesia. After thoracotomy, hearts were extracted, rinsed in sodium chloride, and dried on paper. Organs

were fixed in formalin or ROTI® Histofix (Carl Roth, P087.1) or shock-frozen in liquid nitrogen.

A Visual Sonics Vevo 2100 Imaging System with a 40 MHz transducer was used for two-dimensional guided M-mode echocardiography following isoflurane anesthesia (performed blinded by the echocardiography facility).

Ruxolitinib was dissolved in DMSO (100 μg/μL) and diluted in water containing 0.5% methylcellulose (w/v) and 0.1% Tween 80 for administration of 75 mg/kg ruxolitinib twice daily by oral gavage (according to Heine et al, 2013). The vehicle group was treated with corresponding amount of DMSO in water containing 0.5% methylcellulose (w/v) and 0.1% Tween 80. Conditional Jak1 KO was induced in 8-week-old mice with tamoxifen-supplemented chow, according to Meng et al (2018).

## RNA extraction, RT-qPCR, RNA sequencing, and NanoString RNA analysis

To measure gene expression levels, NRVMs were lysed in 500 μL (12-well) RNAzol RT Reagent (Molecular Research Center Inc, RN 190). For the measurement of gene expression levels, NRVMs or powdered mouse hearts were lysed in 500 μL (12-well) or 1 mL (30 mg heart powder) RNAzol RT Reagent (Molecular Research Center Inc, RN 190). CDNA synthesis was performed with iScript cDNA synthesis kit (Bio-Rad, 1,708,840) with 100–200 ng RNA. RT-qPCR was then performed with TaqMan gene expression assays (Thermo Fisher Scientific; Psmd1 - Rn01400483_m1, Jak1 - Mm00600614_m1, Jak2 - Mm01208489_m1, Stat1 - Mm01257286_m1, Stat3 - Mm01219775_m1, Tyk2 - Mm00444469_m1, 18 s - Hs03003631_g1) and SsoAdvancedTM Universal Probes Supermix (BioRad #1725281). RNA sequencing was performed as described before (Singh et al, 2021) by the CCHMC sequencing core facility with RNA polyA-stranded library preparation, paired-end 75 bp sequencing conditions, and 20 M reads per sample. Three samples (technical replicates) were used from 3 preparations (biological replicates). NanoString RNA analysis was performed and analyzed by the UKE NanoString facility using nCounter Gene Expression Assay.

## Protein extraction, Western blot, and co-immunoprecipitation

For Western blot analysis, cells were harvested in CelLytic™ M (Sigma-Aldrich, C2978, soluble fraction) or a SDS-based lysis buffer (3% SDS, 30 mM Tris base, pH 8.8, 5 mM EDTA, 30 mM NaF, 10% glycerol, 1 mM DTT, crude or sarcomere- and membrane-enriched fraction, insoluble fraction) with protease inhibitor cocktail (cOmplete™ mini, EDTA-free, Roche, 11836170001 and PhosSTOP™, Roche, PHOSS-RO). Mouse tissue samples were powdered; and about 30 mg of powder were dissolved in 180 μL CelLytic M with protease inhibitor cocktail and homogenized by using a tissue lyzer (2 × 30 s at 20 Hz) and centrifuged at 4 °C, full speed for 30 min in a table-top centrifuge. The pellet of the first step was homogenized in 180 μL SDS buffer and centrifuged at room temperature, full speed, for 15 min in a table-top centrifuge. The co-immunoprecipitation was performed according to Thottakara et al (2015). Afterward, 8–25 μg protein with 6x Laemmli buffer was loaded onto 4–15% or 4–20% Mini-PROTEAN TGX™ Precast Protein Gels (Bio-Rad, 4561084) or 8 or 12% self-casted acrylamide/bisacrylamide (29:1) gels. After

## The paper explained

### Problem

Protein aggregation disrupts cellular function and underlies severe degenerative disorders collectively known as proteinopathies, including desmin-related (cardio)myopathy (DRM). Current treatment options for cardiac proteinopathies are limited to symptomatic management, as no disease-specific therapies are available. This significant unmet clinical need underscores the urgency of developing targeted strategies that directly reduce pathological protein aggregation in the heart.

### Results

We investigated the FDA-approved Janus kinase (JAK) inhibitor ruxolitinib as a potential therapeutic approach for cardiac proteinopathy. Ruxolitinib cleared pre-existing protein aggregates in cardiomyocytes by enhancing ubiquitin–proteasome system activity (UPS). In engineered heart tissues, ruxolitinib demonstrated low cardiotoxicity. In a mouse model of cardiac proteinopathy, treatment suppressed pathological JAK signaling and prevented the development of cardiac dysfunction.

### Impact

Our findings show that FDA-approved JAK inhibitors, such as ruxolitinib, effectively reduce protein aggregate burden in cardiac proteinopathy models by inhibiting JAK signaling and augmenting UPS-mediated degradation. These results highlight the potential for repurposing JAK inhibitors as disease-modifying therapies for degenerative disorders characterized by αB-crystallin-containing protein aggregates.

electrophoresis, the proteins were transferred to nitrocellulose or PVDF membranes with Bio-Rad Tris/Glycine buffer or self-made transfer buffer (25 mM Tris base, 190 mM glycine, 20% methanol, pH 8.3) and stained with the following antibodies: JAK1 (R&D Systems, MAB4260); STAT3 (Cell Signaling Technology, 12640); P-STAT3 (Cell Signaling Technology, 9145S); GAPDH (HyTest, 5G4 or Sigma-Aldrich, P0067); ACTN2 (Sigma-Aldrich, A7811); GFP (Santa Cruz Biotechnology, sc-9996); LC3B (Cell Signaling Technology, 2775); p70 S6K (Cell Signaling, 34475); CRYAB (Enzo Life Sciences, ADI-SPA-222-F or Santa Cruz, sc-137129); desmin (Sigma-Aldrich, SAB5600054); FLAG (Sigma-Aldrich, F7425); ubiquitin (Enzo Life Sciences, BML-PW 8810). Western blots were detected with LI-COR Biosciences Odyssey, Biorad ChemiDoc, or Vilber Fusion FX7. Quantifications were done with Image Studio (LICOR), Image Lab (Bio-Rad), or Bio-1D (Vilber) software.

### Fluorescence imaging

NRVMs or hiPSC-CMs were fixed in 4% PFA for 15–20 min at 4 °C, permeabilized for 15 min in PBS/0.5% Triton X100, and stained for the cardiomyocyte marker TNNI3 (troponin I, cardiac 3; Millipore, MAB1691) or ACTN2 (Sigma-Aldrich, A7811), nuclei (DAPI, Thermo Fisher Scientific, D1306) or phalloidin (phalloidin-Alexa Fluor 633 Thermo Fisher Scientific A22284) in blocking solution (PBS, 1% BSA, 0.1% Tween 20, 0.05% sodium azide) and mounted with ProLongTM Gold antifade (Thermo Fisher Scientific #P36930) for LabTeks or nanopatterned coverslips or kept (well plates) in PBS for imaging. Mouse hearts were fixed in formalin or ROTI® Histofix (Carl Roth, P087.1), embedded in paraffin, sliced and stained with for the cardiomyocyte marker TNNI3 (Millipore, MAB1691), nuclei (DAPI) and CRYAB

(Stressgen, SPA-222). Slides or plates were imaged using a Nikon Eclipse Ti inverted microscope equipped with an Andor Zyla-4.2 sCMOS camera and using a Plan Apo λ 20x/0.75 objective lens or a Zeiss Axio Observer.Z1/7 microscope LSM800 with a Plan-Apochromat 20x/0.80 objective lens. For live cell imaging, cells were stained with SiR-actin (Spirochrome, SC001) or SPY-tubulin (Spirochrome, SC503) and imaged with a Nikon Eclipse Ti2-E inverted microscope equipped with an Okolab incubation chamber or Nikon BioStation IM-Q time-lapse imaging system with a 40x/0.80 objective lens. Whole images were quantified for aggregates within cardiomyocytes or F-actin-positive cells with NIS Elements software or ImageJ by setting a threshold for each channel, using the default method, and defining the cardiomyocyte area as a region of interest (ROI).

## Statistical analysis

Statistical tests were performed with GraphPad Prism 8. The applied test and $p$-value are indicated in the figure or figure legend. Sample sizes are indicated in the figure legends. A $p$-value $< 0.05$ was considered significant.

## Data availability

This study includes no data deposited in external repositories.

The source data of this paper are collected in the following database record: biostudies:S-SCDT-10_1038-S44321-026-00411-x.

## Peer review information

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

## Acknowledgements

This research was supported by the DZHK (German Centre for Cardiovascular Research) no. 81X3710111 (SRS), the Ernst-and-Berta-Grimmke-Stiftung no. 6/22 (SRS), UKE "Close the Gap" (SRS), Fondation Leducq no. 11CVD04 (LC, JR), Fondation Leducq no. 20CVD01 (LC), DAAD (JJKM), and RISE (JJKM). The authors thank Dr. Ingke Braren (UKE Vector Facility) for virus productions, Kristin Hartmann (UKE Experimental Histo-Pathology Facility) for histology, Kritton Shay-Winkler (CCHMC) for NRVM preparations, Thomas Schulze, Birgit Klampe, and Prof. Dr. Arne Hansen for help with hiPSC-CMs, the CCHMC Sequencing Facility for RNA sequencing, CCHMC Cardiovascular Imaging for echocardiography, and Melvil Roi for literature search on E3 ligases.

## Author contributions

**Erda Alizoti**: Validation; Investigation; Visualization; Methodology; Writing—original draft; Writing—review and editing. **Leonie Ewald**: Investigation; Methodology; Writing—review and editing. **Simona Parretta**: Investigation; Writing—review and editing. **Jacob J K March**: Data curation; Investigation; Visualization; Writing—review and editing. **Moritz Meyer-Jens**: Investigation; Methodology; Writing—review and editing. **Ellen Orthey**: Investigation; Methodology; Writing—review and editing. **Christian Conze**: Investigation; Methodology; Writing—review and editing. **Lucie Carrier**: Funding acquisition; Project administration; Writing—review and editing. **Jeffrey Robbins**: Conceptualization; Funding acquisition; Project administration; Writing—review and editing. **Sonia R Singh**: Conceptualization; Resources; Data curation; Formal analysis; Supervision; Funding acquisition; Validation; Investigation; Visualization; Methodology; Writing—original draft; Project administration; Writing—review and editing.

Source data underlying figure panels in this paper may have individual authorship assigned. Where available, figure panel/source data authorship is

listed in the following database record: biostudies:S-SCDT-10_1038-S44321-026-00411-x.

## Funding

## Disclosure and competing interests statement

The authors declare no competing interests.

# Expanded View Figures

**Figure EV1.  Ruxolitinib treatment prevents CRYAB^R120G aggregate formation at early and late stages and partially clears pre-existing aggregates.** ▶

(**A**) Ruxolitinib concentration-response curve on phosphorylated STAT3 levels in NRVMs. NRVMs were treated with indicated concentrations of ruxolitinib or 0.1% DMSO. Western blot of protein extracts from treated NRVMs was stained with antibodies directed against phosphorylated (P-) STAT3 and STAT3. (**B, C**) NRVMs were transduced with AdV5-CMV-CRYAB^R120G-GFP and treated with either 3 μM ruxolitinib or DMSO on the indicated day. Medium change with ruxolitinib or DMSO was performed every other day. (**B**) Quantification of aggregates in cardiomyocytes with NIS Elements software. Data were obtained from 2 independent NRVM preparations (prep) with 3 wells per prep, 7–8 images per well and are depicted as mean ± SEM, one-way ANOVA, Dunnett's post-hoc analysis for the same time points or unpaired Student's t-test for comparison of start and end of the experiment within one group. (**C**) Representative IF images. Aggregates are depicted in magenta (CRYAB^R120G-GFP), cardiomyocytes in yellow (anti-cardiac troponin I), and nuclei in blue (DAPI). Scale bar = 100 μm.

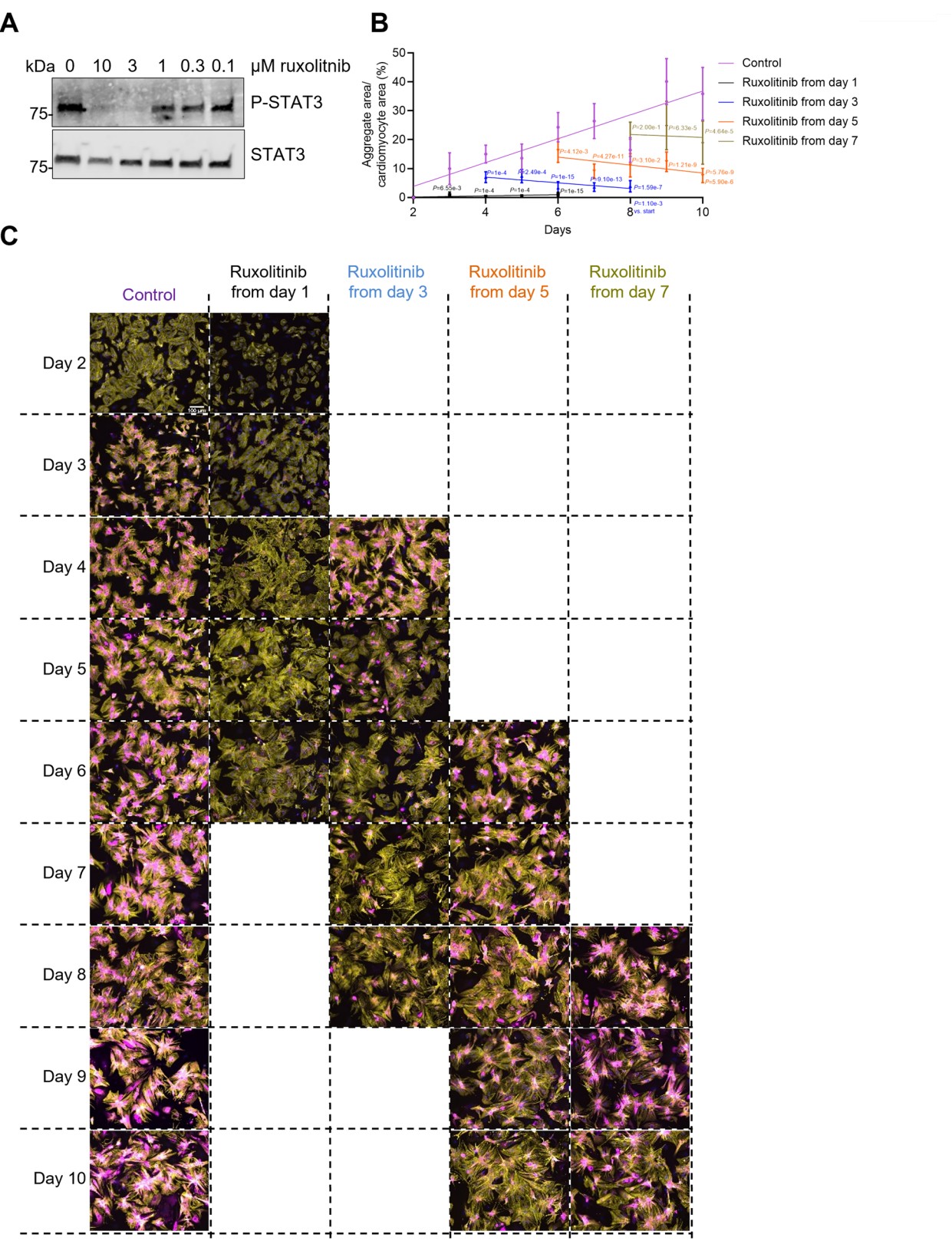

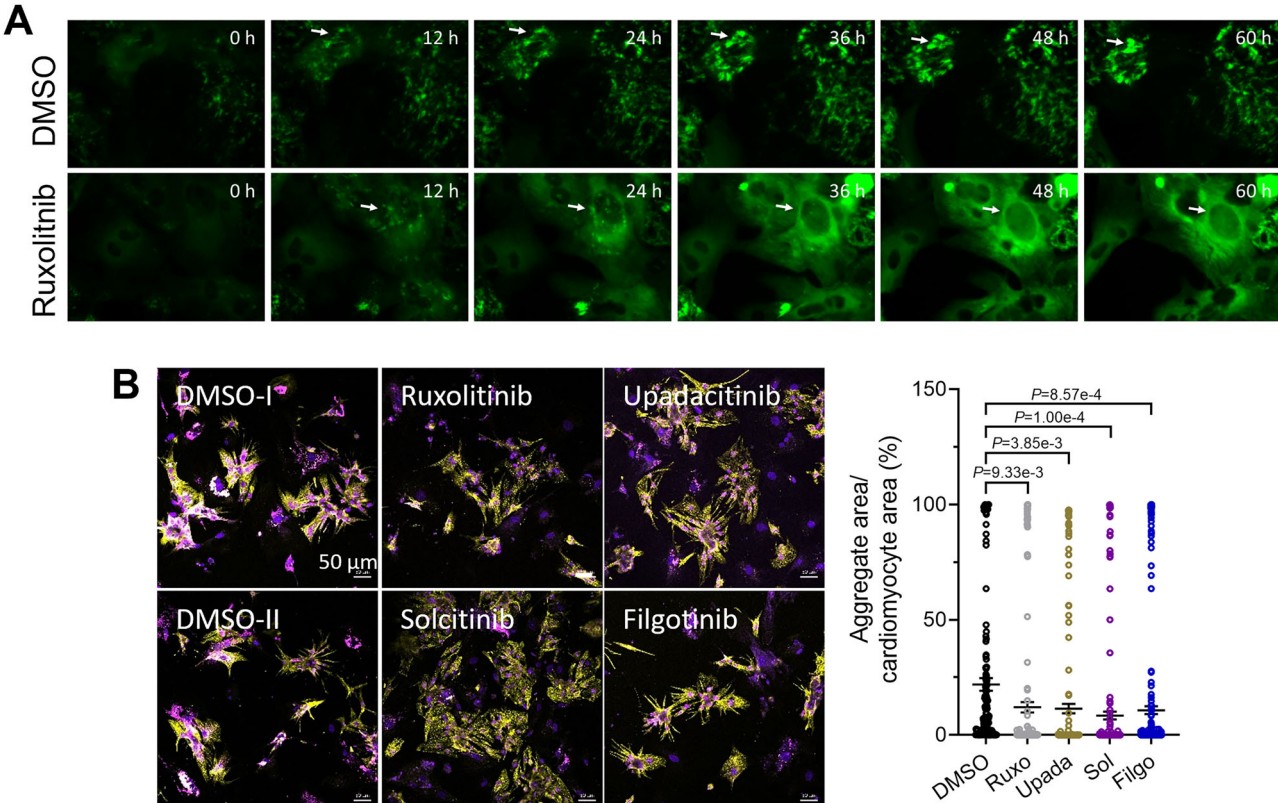

**Figure EV2.  Ruxolitinib and selective JAK1 inhibitors reduce CRYAB[R120G] aggregates in hiPSC-CMs.**

(A) Live cell imaging of hiPSC-CMs transduced with AdV5-CMV-CRYAB[R120G]. Representative fluorescence images of videos from hiPSC-CMs treated with 3 µM ruxolitinib or 0.1% DMSO were taken with a Nikon Biostation IM-Q time-lapse with a Nikon Biostation IM-Q time lapse imaging system (widefield). Aggregates are depicted in green (CRYAB[R120G]-GFP). Arrows mark growing or dissolving aggregates. Corresponding videos can be found in online supplements. (B) Representative immunofluorescence images of NRVMs treated with 3 µM ruxolitinib or 1 µM upadacitinib, solcitinib or filgotinib or 0.1% DMSO, transduced with AdV5-CMV-CRYAB[R120G]-GFP and fixed after 4–6 days. Aggregates are depicted in magenta (CRYAB[R120G]-GFP), cardiomyocytes in yellow (anti-cardiac troponin I), and nuclei in blue (DAPI). Quantification of aggregates in cardiomyocytes (NRVMs) with ImageJ software. Scale bar = 50 µm. Data were obtained from 2 independent NRVM preparations with at least 4 wells per condition and 5–8 images per well and are depicted as mean ± SEM, with $p$-values obtained with the one-way ANOVA and Dunnett's post-hoc analysis.

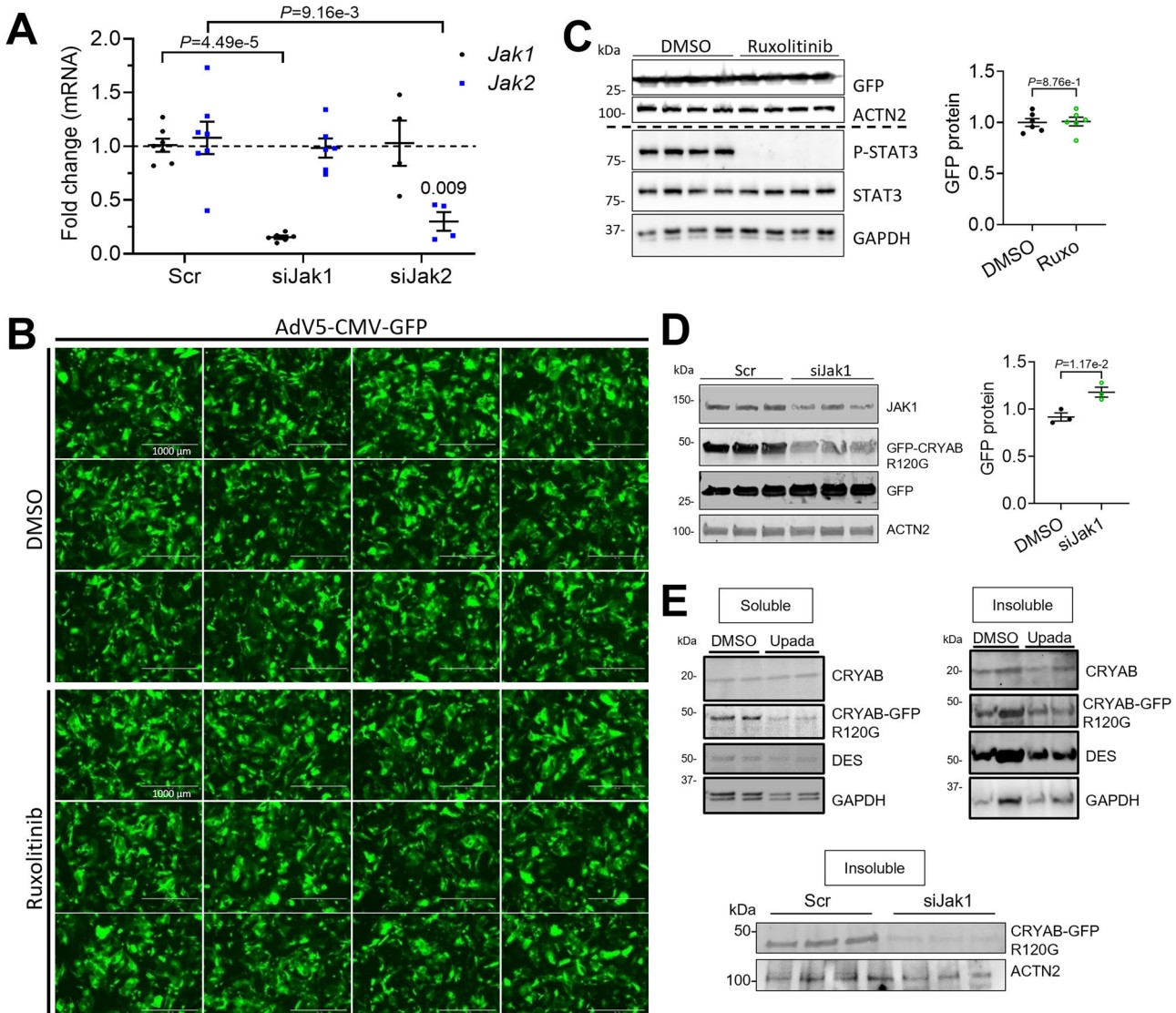

**Figure EV3. Ruxolitinib or siJak1 treatment does not result in lower GFP levels in immunofluorescence or blot.**

NRVMs were transduced with AdV5-CMV-GFP (**A–C**) and AdV5-CMV-CRYAB[R120G]-GFP (**C**) and treated with either 3 µM ruxolitinib or DMSO directly after transduction (**A, B**) or transfected with siRNA targeting *Jak1* or *Jak2* (siJak1, siJak2), or scramble siRNA (scr) directly before transduction (**C**). Medium change with ruxolitinib or DMSO was performed every other day and cells were fixed or harvested 5 days after transduction. (**A**) mRNA levels of GFP after siRNA treatment. (**B**) IF images. Scale bar = 1000 µm. (**C–E**) Western blots of protein extracts from treated NRVM were stained with antibodies directed against indicated proteins. Quantification was performed with Image Lab (**B**) or Image Studio (**C**) software. Data are depicted as mean ± SEM, and *p*-value was obtained with the unpaired Student's t-test. ns, non-significant.

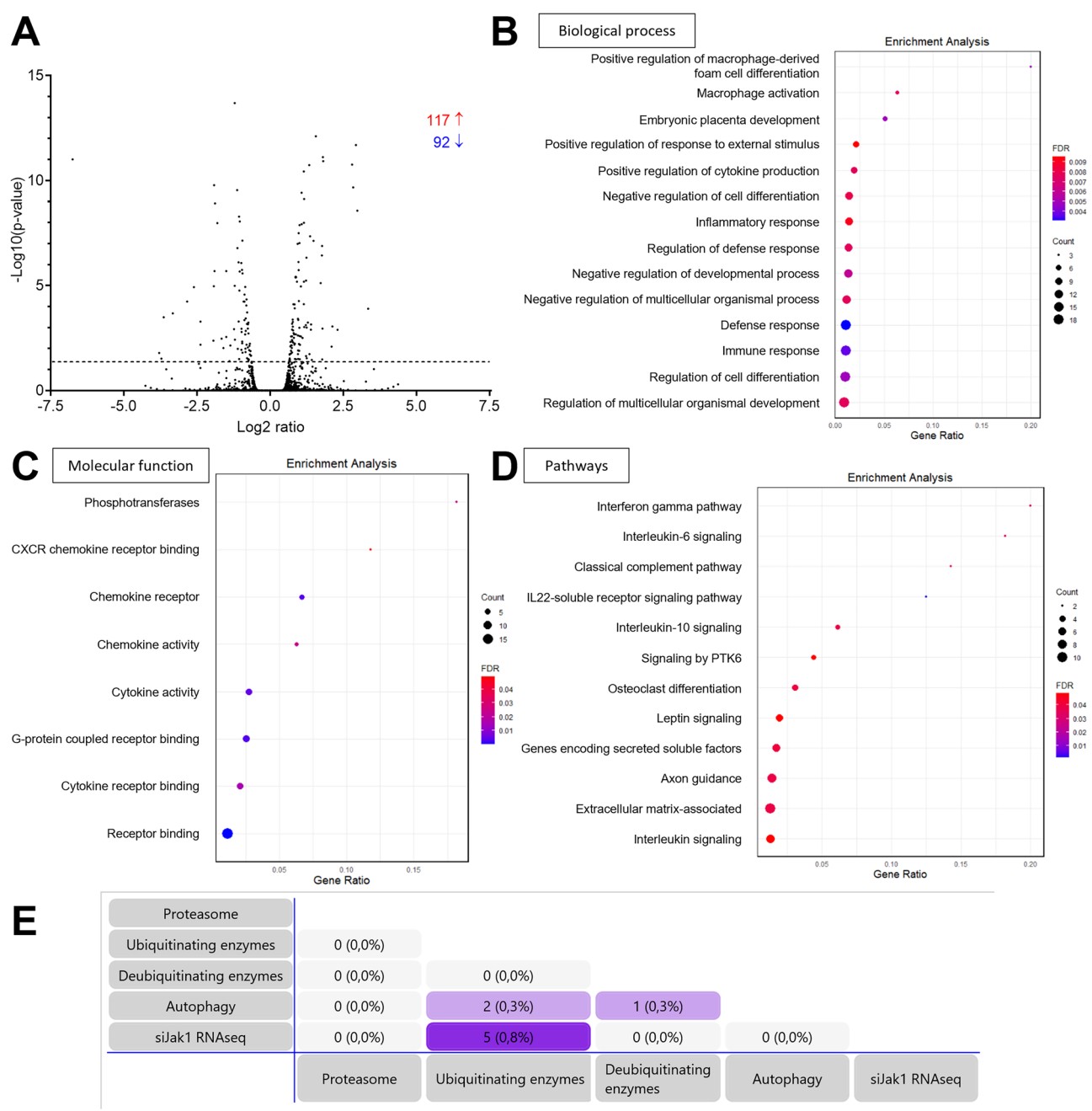

**Figure EV4.  Effects of *Jak1* knockdown on RNA levels in NRVMs.**

NRVMs were transfected with 100 nM siRNA targeting *Jak1* or scramble siRNA and extracted RNA was subjected to paired-end RNA sequencing. Data were obtained from 3 samples of 3 independent NRVM preparations ($n = 3$; each replicate was pooled from 3 samples of 1 NRVM preparation). (**A**) Volcano plots show the $-\log10$ of *P*-value vs. the magnitude of change (log2 ratio) of mRNA levels in siJak1/scr. Differential gene expression analysis was used. Dot plots of enrichment analysis of (**B**) biological processes, (**C**) molecular function, and (**D**) pathways. (**E**) Mapping of significantly up- or down-regulated RNAs to gene identifiers related to proteasome, ubiquitinating enzymes, deubiquitinating enzymes and autophagy.

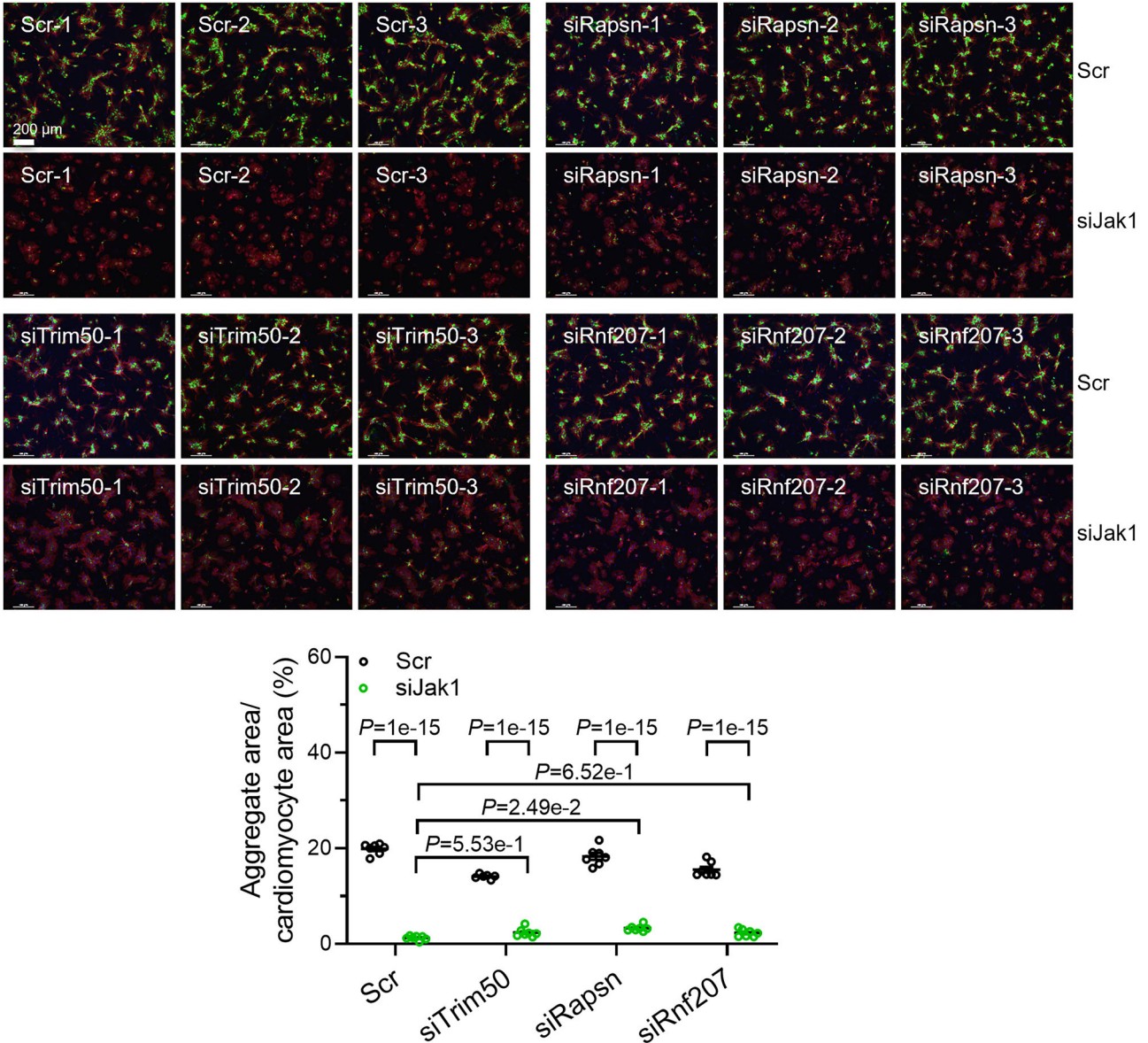

**Figure EV5. Knockdown of E3 ligase *Trim50*, *Rapsn* or *Rnf207* has no major effect on CRYAB[R120G] aggregates after *Jak1* knockdown.**

NRVMs transfected with 100 nM siTrim50, siRapsn, siRnf207 or scramble siRNA (scr), transduced with AdV5-CMV-CRYAB[R120G], treated with 3 µM ruxolitinib (ruxo) or DMSO and fixed after 5 days. Representative immunofluorescence images. Scale bar = 200 µm. Aggregates are depicted in green (CRYAB[R120G]-GFP), cells in red (anti-cardiac troponin I) and nuclei in blue (DAPI). Quantification of aggregates in cardiomyocytes with NIS Elements software. Data were obtained from 1 NRVM preparation with 2 wells per condition and 3 images per well, and are depicted as mean ± SEM, and *p*-values were obtained with the two-way ANOVA with Tukey's multiple comparisons post-hoc analysis. Dots represent images.

