## [Peer Review File · EMBO Molecular Medicine]

JAK1/2 inhibitor ruxolitinib reduces aggregates in cardiac proteinopathy

Erda Alizoti, Leonie Ewald, Simona Parretta, Jacob March, Moritz Meyer-Jens, Ellen Orthey, Christian Conze, Lucie Carrier, Jeffrey Robbins, and Sonia Singh

Corresponding author: Sonia Singh (s.singh@uke.de)

Review Timeline:

Submission Date:	17th Feb 25
Editorial Decision:	10th Mar 25
Revision Received:	11th Aug 25
Editorial Decision:	2nd Sep 25
Revision Received:	2nd Dec 25
Editorial Decision:	12th Feb 26
Revision Received:	9th Mar 26
Accepted:	9th Mar 26

Editor: Jingyi Hou

Transaction Report:

10th Mar 2025

Dear Dr. Singh,

Thank you for submitting your work to EMBO Molecular Medicine. We have now heard back from the three reviewers who agreed to evaluate your manuscript.

As you will see below, while Referees #1 and Referee #3 are generally more supportive, Referee #2 is more reserved and highlights limitations such as the mechanisms, the modest effect of JAK1 or STAT3 inhibition on CRYABR120G cardiomyopathy, and the compromised conceptual novelty due to a previous in vitro screening paper from 2017. Additionally, there are overlapping concerns from the other two reviewers regarding the mechanisms and certain technical aspects of the study.

During our pre-decision cross-commenting process (in which the referees are given a chance to make additional comments, including on each other's reports), Referee #2 reiterated these concerns, while Reviewer #3 still thinks that the study has a high translational potential. I have included these comments following their reports. Considering the balance of the review, we think we can give you a chance to revise the manuscript, with the understanding that all the raised technical issues need to be addressed, and the mechanism needs to be strengthened.

As you may already know, our editorial policy allows in principle a single round of major revision, so it is essential to provide responses to the reviewers' comments that are as complete as possible. Please feel free to contact me in case you would like to discuss in further detail any of the issues raised by the reviewers.

Please also contact us as soon as possible if similar work is published elsewhere. If other work is published, we may not be able to extend the revision period beyond three months.

I look forward to receiving your revised manuscript.

Use this link to login to the manuscript system and submit your revision: <https://embomolmed.msubmit.net/cgi-bin/main.plex>

Sincerely,
Jingyi

Jingyi Hou
Senior Editor
EMBO Molecular Medicine

We require:

- 1) A .docx formatted version of the manuscript text (including legends for main figures, EV figures and tables). Please make sure that the changes are highlighted to be clearly visible.
- 2) Individual production quality figure files as .eps, .tif, .jpg (one file per figure). For guidance, download the 'Figure Guide PDF': (<https://www.embopress.org/page/journal/17574684/authorguide#figureformat>).
- 3) A .docx formatted letter INCLUDING the reviewers' reports and your detailed point-by-point responses to their comments. As

part of the EMBO Press transparent editorial process, the point-by-point response is part of the Review Process File (RPF), which will be published alongside your paper.

4) A complete author checklist, which you can download from our author guidelines (<https://www.embopress.org/page/journal/17574684/authorguide#submissionofrevisions>). Please insert information in the checklist that is also reflected in the manuscript. The completed author checklist will also be part of the RPF.

6) It is mandatory to include a 'Data Availability' section after the Materials and Methods. Before submitting your revision, primary datasets produced in this study need to be deposited in an appropriate public database, and the accession numbers and database listed under 'Data Availability'. Please remember to provide a reviewer password if the datasets are not yet public (see <https://www.embopress.org/page/journal/17574684/authorguide#dataavailability>).

.

- the medical issue you are addressing,

- the results obtained and

- their clinical impact.

This may be edited to ensure that readers understand the significance and context of the research. Please refer to any of our

published articles for an example.

12) Author contributions: You will be asked to provide CRediT (Contributor Role Taxonomy) terms in the submission system. These replace a narrative author contribution section in the manuscript.

13) A Conflict of Interest statement should be provided in the main text.

14) Every published paper now includes a 'Synopsis' to further enhance discoverability. Synopses are displayed on the journal webpage and are freely accessible to all readers. They include a short stand first (maximum of 300 characters, including space) as well as 2-5 one-sentence bullet points that summarize the paper. Please write the bullet points to summarize the key NEW findings. They should be designed to be complementary to the abstract - i.e. not repeat the same text. We encourage inclusion of key acronyms and quantitative information (maximum of 30 words / bullet point). Please use the passive voice. Please attach these in a separate file or send them by email, we will incorporate them accordingly.

Please also suggest a visual abstract to illustrate your article as a PNG file 550 px wide x 300-600 px high.

15) All Materials and Methods need to be described in the main text using our 'Structured Methods' format. According to this format, the Methods section includes a Reagents and Tools Table (listing key reagents, experimental models, software and relevant equipment and including their sources and relevant identifiers) followed by a Methods and Protocols section describing the methods, ideally using a step-by-step protocol format. The aim is to facilitate adoption of the methodologies across labs.

Please download and fill our Reagents and Tools Table template (.docx), which you can find in our author guidelines: <https://www.embopress.org/page/journal/17574684/authorguide#structuredmethods>

When submitting your revised manuscript, please DO NOT include the Reagents and Tools Table in the Methods section of the manuscript but upload it as a separate file choosing the file type "Reagent Table".

***** Reviewer's comments *****

Referee #1 (Remarks for Author):

The manuscript presents an overall well-designed study investigating the potential therapeutic effects of the JAK1/2 inhibitor ruxolitinib in reducing protein aggregates in a model of desmin-related cardiomyopathy (DRM). The findings are novel and potentially impactful, as they suggest a new therapeutic avenue for proteinopathies, which are currently difficult to treat. The study is well-executed, with a combination of in vitro and in vivo experiments, and the data are generally robust. However, the study requires additional experiments and clarifications to fully support the conclusions and address the potential limitations.

Major Concerns:

1. The manuscript suggests that JAK1 inhibition is the primary mechanism by which ruxolitinib reduces CRYAB[R120G] aggregates. However, ruxolitinib is a JAK1/2 inhibitor, and the authors do not fully address the potential contribution of JAK2 inhibition to the observed effects. While the authors show that JAK2 knockdown does not reduce aggregates, this does not entirely rule out a role for JAK2 in the context of ruxolitinib treatment. The authors should provide more direct evidence that JAK1, and not JAK2, is the key player in this process by including experiments with a more selective JAK1 inhibitor (e.g., upadacitinib, filgotinib).
2. Importantly, while the paper convincingly demonstrates the effect of JAK1 inhibition on aggregate clearance, it does not explore the exact molecular pathways linking JAK1-STAT3 signaling to proteasomal degradation in sufficient detail.
3. Additionally, while the authors show that autophagy is not significantly affected by ruxolitinib, they do not explore whether autophagy might play a compensatory role in the absence of UPS activity. This could be particularly relevant in the context of proteinopathies, where both pathways are often implicated.
4. The in vivo data show that ruxolitinib prevents cardiac dysfunction in DRM mice, but the authors do not address whether this is due to a direct effect on cardiomyocytes or an indirect effect (e.g., through reduced inflammation or fibrosis). Given that JAK-STAT signaling is also involved in immune responses, it is important to clarify whether the observed benefits are due to a direct effect on cardiomyocytes or an indirect effect on the cardiac microenvironment.
5. The study focuses on short-term treatment with ruxolitinib, but the long-term effects of JAK1/2 inhibition in the context of DRM are not explored. Given that ruxolitinib is known to have side effects, including immunosuppression and hematological toxicity, it would be important to assess whether long-term treatment is feasible and safe in this context. Additionally, the study should clarify whether ruxolitinib can be effectively used in clinical scenarios, as high doses could lead to hematopoietic toxicity.

Minor Concerns:

1. The discussion section could be expanded to include a more thorough consideration of the limitations of the study. For example, the authors should discuss the potential off-target effects of ruxolitinib and the limitations of using a single DRM model

(CRYAB[R120G]) to draw broader conclusions about proteinopathies.

2. The supplementary data are extensive, but some of the information (e.g., RNAseq data) could be better integrated into the main text or summarized more effectively. For example, the authors could provide a table summarizing the key upregulated and downregulated genes related to UPS and autophagy pathways.
3. Several figures lack clear statistical annotations and Sample sizes.
4. How early in the disease should ruxolitinib be administered to be effective? Data from early vs. late intervention groups would strengthen the translational impact.

Referee #2 (Comments on Novelty/Model System for Author):

The study shows the same results as the 2017 paper by McLendon et al. The in vivo effect of JAK1/STAT3 inhibition upon desmin cardiomyopathy is modest. The underlying mechanisms, such as STAT3-mediated downregulation of E3 ligases, could have been strengthened. The data quality is modest, and data interpretation is shaky.

Referee #2 (Remarks for Author):

The authors show that inhibition of JAK1-STAT3 may alleviate CRYABR120G aggregates by stimulating proteasome-dependent mechanism. This paper follows us the result of McLendon et al (Circ Res 2017).

General:

The study extends the observation in the 2017 paper to in vivo model and shows consistent results. This reviewer finds that the result is short of novel observation. For example, it remains unclear how JAK1-STAT3 negatively regulates proteasome remains unclear.

Specific:

In Figure 4E, siStat3 may not suppress autophagic flux remarkably. The difference between Baf1 and DMSO appears to be comparable between Scr and siStat3. Figure S6 should show the effect of Jak1 inhibition upon LC3 in the presence of BafA1.

Please provide an introduction to E3 ligases studied in S7-9 together with their protein expression and their interaction with CRYABR120G.

The effect of JAK1 or STAT3 inhibition upon CRYABR120G cardiomyopathy is modest. Direct statistical comparison between Veh and Roxo or WT and KO was not conducted.

It would be nice if the authors evaluated whether already established aggregates can be removed by the treatment.

Referee #3 (Comments on Novelty/Model System for Author):

The manuscript entitled "JAK1/2 inhibitor ruxolitinib reduces aggregates in cardiac proteinopathy" addresses a timely and highly pertinent topic related to the mechanisms of protein aggregation associated to DRM. Following up previous data from the group showing that in cardiomyocytes JAK1 knockdown results in lower levels of CRYABR120G aggregation, this study proposes that JAK1 inhibition constitutes a suitable strategy to hinder the accumulation of CRYAB(R120G) aggregates. Furthermore, it is demonstrated that the effect of JAK1 inhibition/ knockdown on CRYABR120G aggregation is mediated by UPS activation. The manuscript is well written and the work very well designed and structured. However, additional experiments are required to strengthen some of the conclusions

Referee #3 (Remarks for Author):

The manuscript entitled "JAK1/2 inhibitor ruxolitinib reduces aggregates in cardiac proteinopathy" addresses a timely and highly pertinent topic related to the mechanisms of protein aggregation associated to DRM. Following up previous data from the group showing that in cardiomyocytes JAK1 knockdown results in lower levels of CRYABR120G aggregation, this study proposes that JAK1 inhibition constitutes a suitable strategy to hinder the accumulation of CRYAB(R120G) aggregates. Furthermore, it is demonstrated that the effect of JAK1 inhibition/ knockdown on CRYABR120G aggregation is mediated by UPS activation. The manuscript is well written and the work very well designed and structured. Despite its biological and clinical relevance, the manuscript presents some weaknesses that need to be addressed and clarified.

. when referring to aggregates dissolution it is stated that the effect of Ruxo is not evident in all cells; in the cells where an aggregates dissolution is observed after Ruxo, is this effect robust leading to aggregates disappearance? Or the cells are differently affected?

. how to distinguish between a decrease in total CRYAB(R120G) levels (fluorescence) and a decrease in protein aggregation? Besides fluorescence levels also the total levels of CRYAB(R120G) determined by westerblot should be presented. Other

techniques to assess protein aggregation should be employed. For example, the ratio soluble vs insoluble CRYAB(R120G), by WB, should be evaluated.

- . The number of replicates in all the experiments should be the same (some experiments have 3 replicates, other 2 and some only have one)
 - . When silencing JAK1 the levels of JAK2 should be presented. The same for JAK2 silencing, where JAK1 amount should be determined.
 - . In Fig.S3C, what is the significance of a decrease in GFP-CRYAB(R120G) amount, after siJAK1, determined by WB? What is the effect of siJAK2 on GFP-CRYAB(R120G) levels?
 - . When investigating the impact of UPS/ 26S inhibition, besides PSMD1 knockdown, also the effect of 20S chemical inhibitors should be considered
 - . In Fig.4, the effect of Ruxo alone on GFPu should be included, as a control.
 - . The effect of Ruxo on GFPu levels, in cells silenced for JAK1/ STAT3 should be assessed.
 - . For UPS studies, the total levels of ubiquitinated proteins should be included
 - . Why in Fig.4C the ACTN2 levels were replaced by ponceau, to assess protein loading?
 - . When evaluating the impact on autophagy, besides LC3, also p62 levels should be determined
 - . In Fig.4E, other loading control than GAPDH should be used, since this protein seems to be affected by experimental conditions (GAPDH levels vary significantly); furthermore, the effect of siSTAT3 on Tor treated cells seems very robust. How to explain this result?
 - . What was previously shown that Desmin aggregates are ubiquitinated and degraded by chaperone-mediated autophagy. Thus, this study should be complemented with data evaluating the impact of Ruxo and JAK1 silencing in CMA and Desmin degradation.
 - . In Fig.6, how are the levels of JAK1 in the NTG and CRYAB(R120G) animals?
 - . In animal experiments, also structural changes at a sarcomeric level, assessed by TEM, should be included (given the impact at Desmin); also the relative levels of Desmin soluble vs insoluble and total amount of ubiquitin conjugates (to assess UPS activity) should be included.
- In Fig.8, phosphorylated-STAT3 levels and the relative levels of CRYAB(R120G) soluble vs insoluble should be considered.

***** Pre-decision cross-commenting *****

Referee #2

The impact of this work is limited. The in vivo data in Figs. 7 and 8 show a minimum improvement of LV function, and it is not evaluated with statistics comparing drugs with and without drugs. R120G does not have any LV dysfunction in Fig. 7C, which indicates inappropriate experimental conditions. The data regarding the involvement of E3 ligase is very preliminary. The essence of other data is already shown in 2017. The autophagy assessment is poorly conducted.

Referee #3

the paper has an high translational potential, even if the effects of JAK1-STAT3 impairment are not very robust. Mechanistic insights about the impact on UPS are missing, and autophagy activity is not properly addressed. A comment raised by Ref#1 about the effect of Ruxo being direct or indirect is very pertinent and needs to be considered, namely the effect at the inflammation and fibrosis level. If the authors can address all these issues and conduct a significant amount of new experiments, as requested by the Referees, the robustness, quality and impact of the work (and conclusions) can significantly increase

Authors: We thank the editor and reviewers for the opportunity to submit our revised manuscript, the critical review of our manuscript, and the thoughtful comments. We have carefully addressed all of the comments in the point-by-point answer below and believe these revisions have significantly enhanced the quality of our manuscript. The updated figure parts are highlighted in the figure files. Please find a detailed explanation of the changes below.

***** Reviewer's comments *****

Referee #1 (Remarks for Author):

The manuscript presents an overall well-designed study investigating the potential therapeutic effects of the JAK1/2 inhibitor ruxolitinib in reducing protein aggregates in a model of desmin-related cardiomyopathy (DRM). The findings are novel and potentially impactful, as they suggest a new therapeutic avenue for proteinopathies, which are currently difficult to treat. The study is well-executed, with a combination of in vitro and in vivo experiments, and the data are generally robust. However, the study requires additional experiments and clarifications to fully support the conclusions and address the potential limitations.

Authors: Thank you for your appreciation of our work and for bringing to our attention the justified concerns. We have performed several additional experiments to support the conclusions and addressed potential limitations.

Major Concerns:

1. The manuscript suggests that JAK1 inhibition is the primary mechanism by which ruxolitinib reduces CRYAB[R120G] aggregates. However, ruxolitinib is a JAK1/2 inhibitor, and the authors do not fully address the potential contribution of JAK2 inhibition to the observed effects. While the authors show that JAK2 knockdown does not reduce aggregates, this does not entirely rule out a role for JAK2 in the context of ruxolitinib treatment. The authors should provide more direct evidence that JAK1, and not JAK2, is the key player in this process by including experiments with a more selective JAK1 inhibitor (e.g., upadacitinib, filgotinib).

Authors: This is a valid concern. We have performed experiments with the more selective JAK1 inhibitors upadacitinib and filgotinib. Both inhibitors resulted in lower CRYAB R120G aggregate in NRVMs and hiPSC-CMs (Figures 2C, D, EV2B). Assessment of CRYAB protein levels in soluble and insoluble fractions with Western blot also revealed lower CRYAB R120G levels with upadacitinib treatment (Figure EV3E). Together with the finding that siJak1 does not affect *Jak2* levels (Figure EV3A) and shows the same effects as ruxolitinib treatment, we believe that JAK1 is the key player in this process.

2. Importantly, while the paper convincingly demonstrates the effect of JAK1 inhibition

on aggregate clearance, it does not explore the exact molecular pathways linking JAK1-STAT3 signaling to proteasomal degradation in sufficient detail.

Authors: The JAK-STAT pathway is an important regulator of gene expression. We found that *Jak1* knockdown led to higher expression of the E3 ligase Asb2. Asb2 β has been shown previously to target desmin for proteasomal degradation (doi.org/10.1016/j.yjmcc.2015.08.020). We added experiments with wildtype (WT) and mutant (L595A) Asb2 β . The mutant form is missing the ligase activity and thus impairs degradation of its substrates. We validated in our model that desmin accumulates if mutant Asb2 β is overexpressed (Figure 6B and C). We found that CRYAB accumulates if the proteasome is inhibited (Figure 4E), confirming that it is degraded by the proteasome. In addition, we found that CRYAB also accumulates with the overexpression of mutant Asb2 β (Figure 6B and C) and co-immunoprecipitates with WT and mutant Asb2 β (Figure 6D), suggesting that it is also a substrate of Asb2 β . Therefore, we hypothesize that in our model JAK1 loss-of-function results in higher expression of Asb2 β , which then increasingly targets desmin and CRYAB for proteasomal degradation, thus resulting in fewer aggregates.

3. Additionally, while the authors show that autophagy is not significantly affected by ruxolitinib, they do not explore whether autophagy might play a compensatory role in the absence of UPS activity. This could be particularly relevant in the context of proteinopathies, where both pathways are often implicated.

Authors: We performed an additional experiment with knockdown of proteasomal function (siPsm1, targeting essential proteasomal subunit) in combination with ruxolitinib treatment (Figure 5C) to test if autophagy is activated with ruxolitinib treatment when the UPS is inhibited. However, we observed a rather inhibited autophagic flux after siPsm1 and ruxolitinib treatment (Figure 5C), further indicating that ruxolitinib treatment does not activate autophagy.

4. The in vivo data show that ruxolitinib prevents cardiac dysfunction in DRM mice, but the authors do not address whether this is due to a direct effect on cardiomyocytes or an indirect effect (e.g., through reduced inflammation or fibrosis). Given that JAK-STAT signaling is also involved in immune responses, it is important to clarify whether the observed benefits are due to a direct effect on cardiomyocytes or an indirect effect on the cardiac microenvironment.

Authors: This is a pressing question. Other studies have shown the beneficial effects of ruxolitinib on fibrosis and inflammation in the heart. We believe that ruxolitinib is likely to have additional beneficial effects on our disease model by potentially reducing fibrosis. The occurrence of inflammation has not been shown for this model. However, addressing these questions is part of our future research, because it requires extensive experimental work, such as systematic evaluation of fibrosis and inflammation in our model, and then targeting JAK1 in fibroblasts and immune cells to

uncover their influence on the disease. This is out of the scope of this project. We added a paragraph to emphasize this limitation of this study (add paragraph, lines 422-429).

5. The study focuses on short-term treatment with ruxolitinib, but the long-term effects of JAK1/2 inhibition in the context of DRM are not explored. Given that ruxolitinib is known to have side effects, including immunosuppression and hematological toxicity, it would be important to assess whether long-term treatment is feasible and safe in this context. Additionally, the study should clarify whether ruxolitinib can be effectively used in clinical scenarios, as high doses could lead to hematopoietic toxicity.

Authors: The reviewer raises an important concern. This issue is challenging to address at present, as the appropriate dosage for treating DRM patients remains unclear. A higher dose could lead to more severe side effects, while a lower dose might reduce adverse effects but be less effective. Additionally, although ruxolitinib was approved in 2011, it is still prescribed to a limited patient population, and off-label uses are not well-documented. This complicates the long-term safety assessment of the drug. Ruxolitinib is approved for long-term use in conditions like myelofibrosis and polycythemia vera, where it can be used for several years. However, due to its immunosuppressive effects and hematological toxicity, close monitoring for infections and blood count abnormalities is crucial. In the future, other inhibitors, such as those targeting STAT3, may also be of interest. Ultimately, the use of ruxolitinib for DRM may constitute off-label treatment, depending on the severity of the DRM progression and the patient's ability to tolerate the drug. This has been discussed in lines 326-334.

More specific

Minor Concerns:

1. The discussion section could be expanded to include a more thorough consideration of the limitations of the study. For example, the authors should discuss the potential off-target effects of ruxolitinib and the limitations of using a single DRM model (CRYAB[R120G]) to draw broader conclusions about proteinopathies.

Authors: The discussion section was edited for a better emphasis on the limitations of the study, such as potential off-target effects of ruxolitinib and the use of a single DRM model (lines 326-334, 432-433)

2. The supplementary data are extensive, but some of the information (e.g., RNAseq data) could be better integrated into the main text or summarized more effectively. For example, the authors could provide a table summarizing the key upregulated and downregulated genes related to UPS and autophagy pathways.

Authors: The RNAseq data (Figure EV4) will be available as expanded view figures with direct access and view from the text. A table with the main E3 ligases regulated was expanded for function and potential interaction with CRYAB and is now available as Table EV2. Apart from that, there were no significantly regulated expression levels related to UPS or autophagy.

3. Several figures lack clear statistical annotations and Sample sizes.

Authors: We improved our statistical annotation and added the minimum sample size to the figure legends. The sample sizes are indicated by the dot plots.

4. How early in the disease should ruxolitinib be administered to be effective? Data from early vs. late intervention groups would strengthen the translational impact.

Authors: This is a pertinent question. We added a paragraph in the discussion to address this (lines 408-416). Excitingly, we found that ruxolitinib can clear pre-existing aggregates in rat and human cardiomyocytes, raising the question if it could reverse the disease to a certain extent. We intentionally started our experiments in mice at a late time point, at 5 months of age for ruxolitinib treatment and at 2 months of age for induced *Jak1* knockout, to explore its translational impact. They revealed that targeting JAK1 after formation of aggregates can still prevent a drop in cardiac function, suggesting that a relatively late start of treatment would still be of significant benefit. Interestingly, the ruxolitinib treatment at 5 months and the *Jak1* knockout at 2 months had similar effects on cardiac function and aggregate load (compare Figures 8 and 9), leading to the assumption that a late intervention may be similarly effective as an early intervention. However, a systematic evaluation would be necessary to answer this question completely.

Referee #2 (Comments on Novelty/Model System for Author):

The study shows the same results as the 2017 paper by McLendon et al. The in vivo effect of JAK1/STAT3 inhibition upon desmin cardiomyopathy is modest. The underlying mechanisms, such as STAT3-mediated downregulation of E3 ligases, could have been strengthened. The data quality is modest, and data interpretation is shaky.

Referee #2 (Remarks for Author):

The authors show that inhibition of JAK1-STAT3 may alleviate CRYABR120G aggregates by stimulating proteasome-dependent mechanism. This paper follows us the result of McLendon et al (Circ Res 2017).

General:

The study extends the observation in the 2017 paper to in vivo model and shows consistent results. This reviewer finds that the result is short of novel observation. For example, it remains unclear how JAK1-STAT3 negatively regulates proteasome remains unclear.

Authors: We respectfully disagree with the comment that the results are short of novel observation. The study of McLendon et al, 2017 identified JAK1 as a target in cell-based high-throughput screening as an effector of CRYAB R120G protein aggregation. Moreover, it showed that siJak1 resulted in fewer CRYAB R120G aggregates and lower GFPu levels in NRVMs. These data were novel and interesting, but very limited for translational impact. In our study, for clinical relevance, we used the approved JAK inhibitor ruxolitinib in rat and human cardiomyocytes, showed that it not only prevented but cleared pre-existing aggregates dependent on the E3 ligase Asb2 β and UPS function, and completely prevented a drop in cardiac function. The effect size on cardiac function and aggregates is comparable to previous studies (e.g. 10.1172/JCI70877), which on the other hand included genetic intervention from birth and not from a late time point as in this study. The data on the mechanism have been strengthened. We now show that Asb2 β binds to CRYAB (Figure 6D), that CRYAB is degraded by the proteasome (Figure 4E), and that inactive Asb2 β leads to accumulation of CRYAB and desmin (Figure 6B and C), indicating that it targets both CRYAB and desmin (doi.org/10.1016/j.yjmcc.2015.08.020) for proteasomal degradation, thus leading to fewer protein aggregates.

Specific:

In Figure 4E, siStat3 may not suppress autophagic flux remarkably. The difference

between Baf1 and DMSO appears to be comparable between Scr and siStat3. Figure S6 should show the effect of Jak1 inhibition upon LC3 in the presence of BafA1.

Authors: All data related to autophagy are now in the new Figure 5. Figure S6 has been removed since it is now redundant with the new Figure 5. We agree with the reviewer that siStat3 treatment does not markedly impair autophagic flux, and it was not our intention to make that point. However, in all previous and new experiments related to autophagy (Figure 5), we found a rather reduced than enhanced autophagic flux, and slightly lower levels of autophagy-related proteins (Figure 5D). Therefore, we would like to point out that autophagy is not responsible for aggregate clearance after ruxolitinib or siJak1 treatment. We have edited the text to make this point clearer (lines 187-188).

Please provide an introduction to E3 ligases studied in S7-9 together with their protein expression and their interaction with CRYABR120G.

Authors: We included an introduction to the E3 ligases and whether an interaction with CRYAB has been reported in Table EV2.

The effect of JAK1 or STAT3 inhibition upon CRYABR120G cardiomyopathy is modest. Direct statistical comparison between Veh and Roxo or WT and KO was not conducted.

Authors: We do not agree. The effect on CRYAB R120G model is comparable to other interventions performed in this model, whereas many of them were genetic interventions from birth (e.g. 10.1172/JCI70877). Our interventions started at 2 or 5 months of age and completely prevented a drop in cardiac function with age. It has to be considered that this is a transgenic CRYAB R120G mouse line with around 5-fold higher levels of CRYAB, leading to a severe phenotype. In addition, hypertrophy is often difficult to reverse in mouse cardiomyopathy models, and many studies show prevention of hypertrophy or use an acute hypertrophy model.

Direct statistical comparison between the R120G groups was added to Figures 8 and 9 for parameters where effects could be seen.

A direct comparison of R120G vehicle and ruxo, and R120G Jak1 WT and KO is now included. The conclusions remain.

It would be nice if the authors evaluated whether already established aggregates can be removed by the treatment.

Authors: We found that established aggregates can be removed by ruxolitinib treatment in rat and human cardiomyocytes (Figures 1 and 2, provided videos). To show aggregate removal in vivo is technically not possible at the moment.

Referee #3 (Comments on Novelty/Model System for Author):

The manuscript entitled "JAK1/2 inhibitor ruxolitinib reduces aggregates in cardiac proteinopathy" addresses a timely and highly pertinent topic related to the mechanisms of protein aggregation associated to DRM. Following up previous data from the group showing that in cardiomyocytes JAK1 knockdown results in lower levels of CRYABR120G aggregation, this study proposes that JAK1 inhibition constitutes a suitable strategy to hinder the accumulation of CRYAB(R120G) aggregates. Furthermore, it is demonstrated that the effect of JAK1 inhibition/knockdown on CRYABR120G aggregation is mediated by UPS activation. The manuscript is well written and the work very well designed and structured. However, additional experiments are required to strengthen some of the conclusions.

Referee #3 (Remarks for Author):

The manuscript entitled "JAK1/2 inhibitor ruxolitinib reduces aggregates in cardiac proteinopathy" addresses a timely and highly pertinent topic related to the mechanisms of protein aggregation associated to DRM. Following up previous data from the group showing that in cardiomyocytes JAK1 knockdown results in lower levels of CRYABR120G aggregation, this study proposes that JAK1 inhibition constitutes a suitable strategy to hinder the accumulation of CRYAB(R120G) aggregates. Furthermore, it is demonstrated that the effect of JAK1 inhibition/knockdown on CRYABR120G aggregation is mediated by UPS activation. The manuscript is well written and the work very well designed and structured. Despite its biological and clinical relevance, the manuscript presents some weaknesses that need to be addressed and clarified.

Authors: We thank the reviewer for the appreciation of our work and thoughtful comments, which improved the manuscript significantly.

. when referring to aggregates dissolution it is stated that the effect of Ruxo is not evident in all cells; in the cells where an aggregates dissolution is observed after Ruxo, is this effect robust leading to aggregates disappearance? Or the cells are differently affected?

Authors: This is an important question. In the cells where dissolution is observed after ruxo, the aggregates do not reappear for the length of the experiment (5 days). As for why cells are differently affected, we hypothesize that there is a certain aggregate or UPS function threshold. When we are beyond this threshold, aggregates cannot be cleared anymore. In our videos it is evident that all cardiomyocytes with smaller aggregates can clear them with ruxolitinib treatment, whereas in cells with larger aggregates, some cells can clear them, while others cannot. This leads us to the assumption that cells with big aggregates have either not reached the threshold yet and the aggregates can still be cleared, or are beyond the threshold, where they cannot be cleared anymore. It is also possible that the aggregation of other proteins

such as desmin plays a role in this context. Within the scope of this study we cannot answer this question, but in a next project we are planning to investigate aggregate formation and clearance in cardiomyocytes more deeply. We have discussed this issue in lines 396-407.

. how to distinguish between a decrease in total CRYAB(R120G) levels (fluorescence) and a decrease in protein aggregation? Besides fluorescence levels also the total levels of CRYAB(R120G) determined by Western blot should be presented. Other techniques to assess protein aggregation should be employed. For example, the ratio soluble vs insoluble CRYAB(R120G), by WB, should be evaluated.

Authors: The aggregates show an extremely bright (oversaturated) fluorescence signal and are very distinct from “non-aggregate” signal, even if there is a decrease in fluorescence. To validate our findings, as suggested, we evaluated soluble and insoluble CRYAB R120G levels by WB. Ruxolitinib led to lower desmin levels in the soluble fraction (probably accessible for UPS in contrast to filament structure, probably no desmin aggregates yet), and lower endogenous CRYAB and CRYAB-GFP R120G levels in the insoluble fraction (probably aggregates; Figure 3B), which confirms our findings observed with IF.

. The number of replicates in all the experiments should be the same (some experiments have 3 replicates, other 2 and some only have one)

Authors: In our experiments, we use a minimum replicate number of 3, except for some concentration-response curves in the supplemental material. For clarification, we added the minimum number of replicates to the figure legends. The numbers are indicated by the dot plots.

. When silencing JAK1 the levels of JAK2 should be presented. The same for JAK2 silencing, where JAK1 amount should be determined.

Authors: The expression levels of *Jak1* and *Jak2* after siJak1 and siJak2 treatment are now presented in Figure EV2A. The siJak1 treatment does not affect *Jak2* and vice versa.

. In Fig.S3C, what is the significance of a decrease in GFP-CRYAB(R120G) amount, after siJAK1, determined by WB? What is the effect of siJAK2 on GFP-CRYAB(R120G) levels?

Authors: The decreased CRYAB-GFP R120G levels after siJak1 knockdown by Western blot indicate fewer aggregates. We now added soluble and insoluble fractions analysis by Western blot for ruxolitinib (Figure 3B), siJak1 and Upadacitinib (Figure EV3). In an experiment with siJak2 knockdown, we unfortunately experienced technical issues with the fractionation and can therefore not add this experiment to

the manuscript. We therefore removed the data regarding siJak2 and aggregate load from the manuscript. The findings regarding the key role of JAK1 remain unaltered.

. When investigating the impact of UPS/ 26S inhibition, besides PSMD1 knockdown, also the effect of 20S chemical inhibitors should be considered

Authors: To test the influence of the UPS system in our model, we need UPS inhibition over 5 days. In different experiments we tried several day UPS inhibition with chemical inhibitors, but were never successful due to toxicity issues. Therefore, we used *Psmc1* or *Psmc1* (not included in the manuscript) knockdown for long-term UPS inhibition and were successful. However, we found that short-term UPS inhibition with epoxomicin resulted in CRYAB R120G accumulation, supporting our *Psmc1* knockdown results. The effect was not as pronounced with epoxomicin as with siPsmc1, because the treatment was only possible the day before harvest due to epoxomicin toxicity.

. In Fig.4, the effect of Ruxo alone on GFPu should be included, as a control.

Authors: The effect of ruxolitinib on GFPu alone was added to Figure 4D. As expected, ruxolitinib led to lower GFPu protein levels, indicating enhanced UPS function.

. The effect of Ruxo on GFPu levels, in cells silenced for JAK1/ STAT3 should be assessed.

Authors: Thank you for making this point. The effect of Ruxo on GFPu in NRVMs silenced for JAK1 was assessed (Figure 4D). Ruxolitinib led to lower GFPu levels in scramble siRNA-treated cells as expected. Ruxolitinib did not lead to any additional lowering of GFPu levels in NRVMs treated with siJak1, suggesting that JAK1 loss of function is responsible for the enhanced UPS function after ruxolitinib treatment.

. For UPS studies, the total levels of ubiquitinated proteins should be included

Authors: We included the total level of ubiquitinated proteins in the soluble and insoluble fraction of Ruxo- or DMSO-treated NRVMs (Figure 4E). The ubiquitinated protein levels were markedly lower with ruxolitinib and epoxomicin treatment compared to the control in the soluble fraction. This result indicates that ruxolitinib treatment enhanced UPS activity, thus leading to fewer ubiquitinated proteins present in the cells.

. Why in Fig.4C the ACTN2 levels were replaced by ponceau, to assess protein loading?

Authors: There was no particular reason. In the experiment, the ACTN2 staining may not have worked due to technical issues. Nonetheless, we replaced the experiment with ACTN2 control (Figure 4C). The conclusions remain the same.

. When evaluating the impact on autophagy, besides LC3, also p62 levels should be determined

Authors: Due to the amount of data from the new experiments the Figure 4 has been separated into two figures. All data related to autophagy can now be found in Figure 5. From our experience, p62 autophagic flux did not give robust results in cardiomyocytes. In addition, alteration of UPS function and the presence of protein aggregates affected the levels of p62, limiting its potential as an autophagy marker in our hands. We included p62 protein levels in the new autophagy experiment (Figure 5C), but did not observe any marked differences on p62 levels or flux with ruxolitinib treatment. In addition, p62 did not accumulate with bafilomycin A1 treatment in the control group, limiting the conclusions for the autophagic flux.

. In Fig.4E, other loading control than GAPDH should be used, since this protein seems to be affected by experimental conditions (GAPDH levels vary significantly); furthermore, the effect of siSTAT3 on Tor treated cells seems very robust. How to explain this result?

Authors: Due to the amount of data from the new experiments the Figure 4 has been separated into two figures. All data related to autophagy can now be found in Figure 5. We added other controls to now Figure 5B, but found GAPDH to be the most suitable one. The overall conclusion on autophagy not being responsible for aggregate clearance after ruxolitinib, siJak1 or siStat3 treatment remains.

Tor is an early-step autophagy activator (mTOR inhibitor), increasing the autophagic flux. In our hands, inhibition or knockdown of JAK1 or STAT3 results in a mild reduction of autophagic activity (Figure 5). This could lead to LC3-II accumulation after siStat3 treatment in combination with torin. Similarly to the fact that LC3-II accumulation is higher after combined bafilomycin and torin treatment than with bafilomycin alone.

. What was previously shown that Desmin aggregates are ubiquitinated and degraded by chaperone-mediated autophagy. Thus, this study should be complemented with data evaluating the impact of Ruxo and JAK1 silencing in CMA and Desmin degradation.

Authors: We evaluated levels of proteins involved in CMA, such as Hsc70 and BAG3, by WB (Figure 5D). Hsc70 and BAG3 protein levels were slightly lower in ruxolitinib- than DMSO-treated cells. Overall, all results related to autophagy indicate a mildly reduced autophagic activity in cells treated with ruxolitinib.

. In Fig.6, how are the levels of JAK1 in the NTG and CRYAB(R120G) animals?

Authors: The levels of JAK1 were higher in CRYAB R120G than in NTG animals, which is in line with our hypothesis that the JAK-STAT pathway is activated in these mice (Figure 5F).

. In animal experiments, also structural changes at a sarcomeric level, assessed by TEM, should be included (given the impact at Desmin); also the relative levels of Desmin soluble vs insoluble and total amount of ubiquitin conjugates (to assess UPS activity) should be included.

In Fig.8, phosphorylated-STAT3 levels and the relative levels of CRYAB(R120G) soluble vs insoluble should be considered.

Authors: Thank you for the important comments. We found the levels of desmin to be higher in the soluble and insoluble fraction of R120G mice compared to NTG controls (Figure 7F). Unfortunately, we did not save any samples for EM from the animal experiments, and repetition of the experiments would require approval from authorities for further animal experiments, which takes around 6 months. Nevertheless, we performed high-resolution microscopy with desmin staining to assess desmin and overall structure (Figures 8H and 9H). We found overall improved sarcomeric desmin structure (indicated by green arrows) with ruxolitinib treatment or *Jak1* knockout and fewer desmin aggregates (indicated by white arrows).

Unfortunately, we experienced technical issues with determination of the ubiquitinated protein levels from mouse heart samples. As expected, the levels of phosphorylated STAT3 were lower in the R120G *Jak1* KO than WT group (Figure 9F). Determination of soluble and insoluble CRYAB revealed lower CRYAB levels in the insoluble fraction of R120G ruxolitinib-treated or *Jak1* KO mice in comparison to the corresponding control group (Figures 8G and 9G), validating the finding from IF of heart sections.

***** Pre-decision cross-commenting *****

Referee #2

The impact of this work is limited. The in vivo data in Figs. 7 and 8 show a minimum improvement of LV function, and it is not evaluated with statistics comparing drugs with and without drugs. R120G does not have any LV dysfunction in Fig. 7C, which indicates inappropriate experimental conditions. The data regarding the involvement of E3 ligase is very preliminary. The essence of other data is already shown in 2017. The autophagy assessment is poorly conducted.

Authors: We disagree on these specific points. The study of McLendon et al, 2017 identified JAK1 as a target in cell-based high-throughput screening as an effector of CRYAB R120G protein aggregation. Moreover, it showed that siJak1 resulted in fewer CRYAB R120G aggregates and lower GFPu levels in NRVMs. These data were novel and interesting, but very limited for translational impact. In our study, for clinical relevance, we used the approved JAK inhibitor ruxolitinib in rat and human cardiomyocytes, showed that it not only prevented but cleared pre-existing aggregates dependent on the E3 ligase Asb2 β and UPS function, and completely prevented a drop in cardiac function in vivo. The effect size on cardiac function and aggregates is comparable to previous important studies (e.g. 10.1172/JCI70877), which on the other hand included genetic intervention from birth and not from a late time point as in this study. The statistics required by the reviewer have been added in the in vivo Figures, and the conclusions remain the same. The data on the mechanism have been strengthened. We now show that Asb2 β binds to CRYAB (Figure 6D), that CRYAB is degraded by the proteasome (Figure 4E), and that inactive Asb2 β leads to accumulation of CRYAB and desmin (Figure 6B and C), indicating that it targets both CRYAB and desmin (doi.org/10.1016/j.yjmcc.2015.08.020) for proteasomal degradation, thus leading to fewer protein aggregates. The autophagy assessment has been expanded according to the reviewer's suggestions (Figure 5). No major effect on autophagic activity was observed with ruxolitinib or siJak1. Please find the specific answers in the point-by-point response.

Referee #3

the paper has an high translational potential, even if the effects of JAK1-STAT3 impairment are not very robust. Mechanistic insights about the impact on UPS are missing, and autophagy activity is not properly addressed. A comment raised by Ref#1 about the effect of Ruxo being direct or indirect is very pertinent and needs to be considered, namely the effect at the inflammation and fibrosis level. If the authors can address all these issues and conduct a significant amount of new experiments, as requested by the Referees, the robustness, quality and impact of the work (and conclusions) can significantly increase.

Authors: We thank the reviewer for the appreciation of our work. The important comment of Referee 1 about the effect of Ruxo being direct or indirect was considered and is now discussed in lines 422-429. We also addressed all issues and conducted a significant amount of new experiments as suggested by the reviewer's, which increased the quality of our manuscript. We added important experiments to strengthen the mechanistic insight on the UPS (Figure 6), and further assessed autophagic activity (Figure 5) as suggested by the referees. Please find the specific answers in the point-by-point response.

2nd Sep 2025

Dear Dr. Singh,

Thank you again for submitting the revised version of your manuscript. We have now received feedback from both reviewers who evaluated your revision. As you will see below, while Reviewer #3 is satisfied with the changes made, Reviewer #1 notes that several important issues raised in the previous round of review remain insufficiently addressed.

In principle, our editorial policy permits only a single round of major revision. However, given the potential translational relevance of your study, we are willing to offer an exceptional opportunity for a second-and final-round of major revision. Please note that this will be the last revision round, so it is essential that all remaining concerns are fully addressed. For this reason, and to avoid potential frustrations, we would strongly advise against returning an incomplete revision.

We consider it critical that you address points #2- #3 raised by Reviewer #1 to strengthen the mechanistic and translational aspects of your study. Regarding point #1, please attempt to address the reviewers' suggestion for in vitro mechanistic validation. However, if mechanistic clarification of how JAK knockdown leads to Asb2 β expression cannot be provided experimentally at this time, we would recommend a thoughtful discussion of potential mechanisms and acknowledging this as a direction for future investigation.

We understand that the additional revisions may require substantial experimentation and analyses with uncertain outcomes. Should you decide that these efforts are not feasible, we would, of course, understand if you choose to pursue publication of your study elsewhere.

If you feel you can satisfactorily deal with these points and those listed by the referees, you may wish to submit a revised version of your manuscript. Please attach a covering letter giving details of the way in which you have handled each of the points raised by the referees. A revised manuscript will be once again subject to review and you probably understand that we can give you no guarantee at this stage that the eventual outcome will be favorable.

I look forward to seeing a revised form of your manuscript.

Kind regards
Jingyi

Jingyi Hou
Senior Editor
EMBO Molecular Medicine

We require:

- 1) A .docx formatted version of the manuscript text (including legends for main figures, EV figures and tables). Please make sure that the changes are highlighted to be clearly visible.
- 2) Individual production quality figure files as .eps, .tif, .jpg (one file per figure). For guidance, download the 'Figure Guide PDF': (<https://www.embopress.org/page/journal/17574684/authorguide#figureformat>).
- 3) A .docx formatted letter INCLUDING the reviewers' reports and your detailed point-by-point responses to their comments. As part of the EMBO Press transparent editorial process, the point-by-point response is part of the Review Process File (RPF), which will be published alongside your paper.
- 4) A complete author checklist, which you can download from our author guidelines (<https://www.embopress.org/page/journal/17574684/authorguide#submissionofrevisions>). Please insert information in the

checklist that is also reflected in the manuscript. The completed author checklist will also be part of the RPF.

6) It is mandatory to include a 'Data Availability' section after the Materials and Methods. Before submitting your revision, primary datasets produced in this study need to be deposited in an appropriate public database, and the accession numbers and database listed under 'Data Availability'. Please remember to provide a reviewer password if the datasets are not yet public (see <https://www.embopress.org/page/journal/17574684/authorguide#dataavailability>).

12) Author contributions: You will be asked to provide CRediT (Contributor Role Taxonomy) terms in the submission system. These replace a narrative author contribution section in the manuscript.

13) A Conflict of Interest statement should be provided in the main text.

14) Please provide a 'Synopsis' to further enhance discoverability. Synopses are displayed on the journal webpage and are freely accessible to all readers. They include a short stand first (maximum of 300 characters, including space) as well as 2-5 one-sentences bullet points that summarizes the paper. Please write the bullet points to summarize the key NEW findings. They should be designed to be complementary to the abstract - i.e. not repeat the same text. We encourage inclusion of key acronyms and quantitative information (maximum of 30 words / bullet point). Please use the passive voice. Please attach these in a separate file or send them by email, we will incorporate them accordingly.

15) All Materials and Methods need to be described in the main text using our 'Structured Methods' format. According to this format, the Methods section includes a Reagents and Tools Table (listing key reagents, experimental models, software and relevant equipment and including their sources and relevant identifiers) followed by a Methods and Protocols section describing the methods, ideally using a step-by-step protocol format. The aim is to facilitate adoption of the methodologies across labs.

Please download and fill our Reagents and Tools Table template (.docx), which you can find in our author guidelines: <https://www.embopress.org/page/journal/17574684/authorguide#structuredmethods>

When submitting your revised manuscript, please DO NOT include the Reagents and Tools Table in the Methods section of the manuscript but upload it as a separate file choosing the file type "Reagent Table".

**** Reviewer's comments ****

Referee #1 (Remarks for Author):

I thank the authors for their detailed rebuttal and the additional experiments performed in response to my comments. The manuscript has certainly improved in several aspects, particularly with the inclusion of selective JAK1 inhibitor data and mechanistic exploration involving Asb2 β .

Nevertheless, I remain concerned that several of my major points have not been fully addressed, and the study in its current form still falls short of the level of rigor and mechanistic clarity expected for EMBO Molecular Medicine.

1. The additional experiments involving Asb2 β add value, but the mechanistic link remains insufficiently developed. The conclusion that JAK1 loss-of-function upregulates Asb2 β and thereby promotes proteasomal degradation is still largely correlative. Stronger mechanistic validation, such as in vitro ubiquitination assays, ubiquitin pulldowns, or rescue experiments, would be required to substantiate this claim. Moreover, it remains unclear how JAK1 knockdown mechanistically leads to increased Asb2 β expression. Without addressing this mechanistic connection, the proposed pathway remains speculative.

2. in vivo mechanism (direct vs. indirect effects): This remains a major gap. The authors acknowledge that ruxolitinib may act via fibrosis or inflammation but dismiss this as "out of scope." Given the known role of JAK/STAT signaling in immune and fibroblast biology, this omission substantially limits the translational impact. Without discriminating between direct cardiomyocyte effects and microenvironmental influences, the in vivo findings cannot be fully attributed to aggregate clearance.

3. Long-term treatment and safety: The rebuttal reiterates the challenges but does not provide new experimental data. Given ruxolitinib's well-documented immunosuppressive and hematologic toxicity, at least some preliminary long-term assessments (even at the preclinical level) are essential before proposing translational implications. The discussion alone does not adequately address this concern.

4. The discussion has been improved but still underplays the limitations of using only the CRYAB(R120G) model to generalize to broader proteinopathies.

Referee #3 (Comments on Novelty/Model System for Author):

The ms is of high technical and scientific quality and can bring important insights towards a future translation to the clinics

Referee #3 (Remarks for Author):

The authors have properly addressed all the issues raised by this reviewer. Some new experiments were added, which increases the robustness of the conclusions and the general impact of the work. This reviewer understands the reasons provided for not performing some few experiments that were requested, which doesn't hamper the publication of the ms in its

present form. Congratulations to the authors for this beautiful work!

***** Reviewer's comments *****

Referee #1 (Remarks for Author):

I thank the authors for their detailed rebuttal and the additional experiments performed in response to my comments. The manuscript has certainly improved in several aspects, particularly with the inclusion of selective JAK1 inhibitor data and mechanistic exploration involving *Asb2* β . Nevertheless, I remain concerned that several of my major points have not been fully addressed, and the study in its current form still falls short of the level of rigor and mechanistic clarity expected for EMBO Molecular Medicine.

Authors: We thank the referee for the thorough review of our manuscript. Please find our point-by-point response below.

1. The additional experiments involving *Asb2* β add value, but the mechanistic link remains insufficiently developed. The conclusion that JAK1 loss-of-function upregulates *Asb2* β and thereby promotes proteasomal degradation is still largely correlative. Stronger mechanistic validation, such as in vitro ubiquitination assays, ubiquitin pulldowns, or rescue experiments, would be required to substantiate this claim. Moreover, it remains unclear how JAK1 knockdown mechanistically leads to increased *Asb2* β expression. Without addressing this mechanistic connection, the proposed pathway remains speculative.

Authors: We hypothesize that, in our model, JAK1 loss-of-function leads to increased *Asb2* expression, which in turn more effectively targets desmin and CRYAB for proteasomal degradation, ultimately resulting in reduced aggregate accumulation. At present, however, we cannot determine whether *Asb2* is regulated directly or indirectly by JAK–STAT signaling, nor which STAT protein might be involved. We acknowledge this as a limitation of the current study and have added potential explanations and a limitation statement to the discussion to clearly address this point (lines 430-435 and lines 488-489). ASB2 is a member of the SOCS (suppressor of cytokine signaling)–box family, many of which are known direct transcriptional targets of STATs (doi: 10.1007/s11515-018-1506-2). JAK–STAT signaling may also modulate *Asb2* expression indirectly, for example, by altering the activity of other transcription factors that regulate the *Asb2* locus. In addition, ASB2 has been reported to promote degradation of JAK2 and JAK3 (PMID: 21969365; PMID: 21119685), suggesting the potential for a feedback loop in which JAK–STAT activity influences *Asb2* expression. Nonetheless, the precise mechanism remains unresolved and will require detailed mechanistic investigation.

We added new experimental data supporting our hypothesis that ASB2 targets CRYAB. We now show that *Asb2* knockdown leads to CRYAB accumulation (Figure 6B), that expression of a ligase-deficient ASB2 β mutant results in CRYAB accumulation that does not further increase upon proteasome inhibition (in contrast to cells expressing wild-type ASB2 β ; Figure 6D), and that ubiquitinated CRYAB levels are reduced in a ubiquitin pulldown assay in the absence or presence of epoxomicin (Figure 6F). Furthermore, CRYAB was significantly enriched in a previous *Asb2* β interactome and ubiquitinome analysis (PMID: 33516941), further supporting our model. This study has now been incorporated into the discussion.

2. in vivo mechanism (direct vs. indirect effects): This remains a major gap. The authors acknowledge that ruxolitinib may act via fibrosis or inflammation but dismiss this as "out of scope." Given the known role of JAK/STAT signaling in immune and fibroblast biology, this omission substantially limits the translational impact. Without discriminating between direct cardiomyocyte effects and microenvironmental influences, the in vivo findings cannot be fully attributed to aggregate clearance.

Authors: We apologize for not describing this clearly enough. The *Jak1* knockout mouse model used in our study is cardiomyocyte-specific, achieved by tamoxifen-induced Cre expression under the *Myh6* promoter. In this model, we observed preserved cardiac function and fewer aggregates, confirming

that the protective effects observed in vivo can be attributed to loss of JAK1 in cardiomyocytes. Ruxolitinib treatment produced effect sizes comparable to the cardiomyocyte-specific *Jak1* knockout, further indicating that the in vivo effects of ruxolitinib are largely mediated through JAK1 inhibition directly within cardiomyocytes. At the same time, if ruxolitinib also exerts effects via modulation of fibrosis or inflammation, this would add to its translational significance. To fully dissect the direct versus indirect contributions of ruxolitinib in vivo, one would need to compare ruxolitinib treatment across mouse models carrying cardiomyocyte-specific, fibroblast-specific, and several immune-cell-specific *Jak1* deletions.

3. Long-term treatment and safety: The rebuttal reiterates the challenges but does not provide new experimental data. Given ruxolitinib's well-documented immunosuppressive and hematologic toxicity, at least some preliminary long-term assessments (even at the preclinical level) are essential before proposing translational implications. The discussion alone does not adequately address this concern.

Authors: According to the reviewer's suggestion to provide some preclinical data, we performed a long-term human induced pluripotent stem cell-derived engineered heart tissue (hiPSC-EHT) experiment to assess ruxolitinib cardiotoxicity (Figure 8). hiPSC-EHTs are an established preclinical model for drug toxicity testing and are recognized by the FDA for this purpose. In patients, ruxolitinib is administered at doses up to 25 mg twice daily, resulting in plasma concentrations of approximately 1 μ M (PMID: 37000342). In our chronic 4-week EHT study, we exposed tissues to 0.3, 1, 3, 10, and 30 μ M ruxolitinib to evaluate potential cardiotoxic effects. We observed only minor changes in contractile force or contraction kinetics, even at concentrations roughly 30-fold higher than those reached in patients. Histological analysis also revealed no major morphological alterations at any concentration, supporting a low cardiotoxicity profile for ruxolitinib. Ruxolitinib has been FDA-approved since 2011, and long-term effects (extending over several years) have been assessed in multiple patient cohorts (e.g. PMID: 28228106; PMID: 31982039; PMID: 40878623). As correctly pointed out by the reviewer, the best-known short-term and long-term adverse reactions in patients include increased susceptibility to infections and reduced blood counts (PMID: 37000342), which is why regular clinical monitoring is recommended during treatment. In-depth toxicology studies in animals were conducted, and a summary is accessible through the FDA webpage (202192Orig1s000). In our study, chronic exposure of engineered heart tissues to ruxolitinib for 4 weeks, even at concentrations considerably higher than those typically reached in vivo, had only minor effects on cardiac tissue structure and function.

4. The discussion has been improved but still underplays the limitations of using only the CRYAB(R120G) model to generalize to broader proteinopathies.

Authors: It is correct that it remains open whether ruxolitinib is also effective on other genetic variants causing protein aggregates, currently limiting its effect to CRYAB aggregates. We now pointed this out clearly in a sentence in lines 489-491 to underline this.

Referee #3 (Comments on Novelty/Model System for Author):

The ms is of high technical and scientific quality and can bring important insights towards a future translation to the clinics

Referee #3 (Remarks for Author):

The authors have properly addressed all the issues raised by this reviewer. Some new experiments were added, which increases the robustness of the conclusions and the general impact of the work. This reviewer understands the reasons provided for not performing some few experiments that were requested, which doesn't hamper the publication of the ms in its present form. Congratulations to the authors for this beautiful work!

12th Feb 2026

Dear Sonia,

Thank you for submitting your revised manuscript. First, please accept my apologies for the delay in the review process. This was due to difficulties in securing reviewers in the period leading up to the holiday season. In addition, the original Reviewer #1, who had agreed to reassess the paper, remained unresponsive despite multiple reminders. We therefore reached out to the original Reviewer #3 and asked them to evaluate your response to Reviewer #1's remaining concerns. We have now received Reviewer #3's report. As you will see, the reviewer is satisfied with the revisions made.

Before we can formally accept your manuscript for publication, we kindly ask that you address the following editorial-level issues:

1. Please add the author information back into the manuscript. It is still blinded in the current version.
2. Please remove the "Authors' Contributions" section from the manuscript file.
3. Please move the section titled "The paper explained" into the main manuscript file.
4. Please remove the list of abbreviations.
5. EV tables: Please remove the table legends from the main manuscript text. Add each legend to the top of its corresponding EV table.
6. Movies: Rename the four movie files as "Movie EV1" through "Movie EV4." Each movie file should include a simple README file containing the legend and be provided as a zipped folder. Update all corresponding callouts in the manuscript accordingly.
7. Source Data :
 - Upload the numerical source data in Excel or CSV format.
 - Please carefully verify that the source data files correspond exactly to the figures presented in the manuscript.
 - Please provide a completed source data checklist, which can be found here: <https://resource-cms.springernature.com/springer-cms/rest/v1/content/27825826/data/v1>
8. During the standard image check, the data integrity officer noted that the blots appear to be over-contrasted. Please review the blots and adjust them if excessive contrast has been applied.
9. Please add the missing callout for Figure 7B. Ensure that Figure 8C is not cited before Figures 7C, 7D, 7E, and 7F.
10. In the reference list, limit the number of listed authors to 10, followed by "et al."
11. Please add an "Acknowledgements" section. Move all funding information to this section. Ensure that the funding details in the manuscript exactly match those entered in the submission system.
12. We updated our journal's competing interests policy in January 2022 and request authors to consider both actual and perceived competing interests. Please review the policy <https://www.embopress.org/competing-interests> and update your competing interests if necessary. Please use the heading "Disclosure statement and competing interests".
13. Remove "data not shown". As per our guidelines on "Unpublished Data", the journal does not permit citation of "Data not shown".
14. Please address the following issues in the figure legends:
 - Please note that the exact p values are not provided in the legends of figures 2B, D; 3A, C, D; 5A, 6A, D; 7C, D, F; 8B, D, G; 9B, EV1 B, EV5
 - Please indicate the statistical test used for data analysis in the legend of figure EV4 A
 - Please note that information related to n is missing in the legends of figures 9B, EV4 A, EV5

I look forward to seeing a revised form of your manuscript as soon as possible.

Sincerely,
Jingyi

Jingyi Hou
Senior Editor
EMBO Molecular Medicine

***** Reviewer's comments *****

Referee #3 (Comments on Novelty/Model System for Author):

The work is highly innovative and strongly supported by high quality data. The authors have properly answered the questions and concerns raised by one the reviewers, during the rebuttal. The explanations provided, namely to what the mechanistic limitations of the study is concerned, are totally acceptable and do not compromise the impact and confidence of the conclusions. Some new data added to this revised version contribute to strengthen the robustness of the main conclusions.

Referee #3 (Remarks for Author):

The authors have properly answered the questions and concerns raised by one the reviewers, during the rebuttal. The explanations provided, namely to what the mechanistic limitations of the study is concerned, are totally acceptable and do not compromise the impact and confidence of the conclusions. Some new data contribute to strengthen the robustness of the main conclusion.

The authors addressed the remaining editorial changes.

9th Mar 2026

Dear Sonia,

We are pleased to inform you that your manuscript is accepted for publication and is now being sent to our publisher to be included in the next available issue of EMBO Molecular Medicine.

You may qualify for financial assistance for your publication charges - either via a Springer Nature fully open access agreement or an EMBO initiative. Check your eligibility: <https://link.springer.com/journal/44321/how-to-publish-with-us>

Sincerely,
Jingyi

Jingyi Hou
Senior Editor
EMBO Molecular Medicine

>>> Please note that it is EMBO Molecular Medicine policy for the transcript of the editorial process (containing referee reports and your response letter) to be published as an online supplement to each paper. If you do NOT want this, you will need to inform the Editorial Office via email immediately. More information is available here: <https://link.springer.com/partners/embo-press/editorial-policies#Peer%20review>